# A universal metasurface antenna to manipulate all fundamental characteristics of electromagnetic waves

Geng-Bo Wu [1,2,7], Jun Yan Dai [3,4,5,7] ✉, Kam Man Shum [1], Ka Fai Chan [1], Qiang Cheng [3,4,5] ✉, Tie Jun Cui [3,4,5] ✉ & Chi Hou Chan [1,2,6] ✉

Metasurfaces have promising potential to revolutionize a variety of photonic and electronic device technologies. However, metasurfaces that can simultaneously and independently control all electromagnetics (EM) waves' properties, including amplitude, phase, frequency, polarization, and momentum, with high integrability and programmability, are challenging and have not been successfully attempted. Here, we propose and demonstrate a microwave universal metasurface antenna (UMA) capable of dynamically, simultaneously, independently, and precisely manipulating all the constitutive properties of EM waves in a software-defined manner. Our UMA further facilitates the spatial- and time-varying wave properties, leading to more complicated waveform generation, beamforming, and direct information manipulations. In particular, the UMA can directly generate the modulated waveforms carrying digital information that can fundamentally simplify the architecture of information transmitter systems. The proposed UMA with unparalleled EM wave and information manipulation capabilities will spark a surge of applications from next-generation wireless systems, cognitive sensing, and imaging to quantum optics and quantum information science.

Electromagnetic (EM) waves, ranging from microwave and terahertz waves to visible light, are the bases of various disciplines ranging from optics[1], telecommunications[2], and material engineering[3,4] to quantum systems[5]. Plane-wave mode with the electric field of $\boldsymbol{E}(\vec{r},t) = \hat{e}A\cos(2\pi f t + \vec{k}\cdot\vec{r} + \varphi)$ is the eigensolution of Maxwell's equation in free space[6,7]. Classical EM waves can be completely characterized by five intrinsic properties: polarization $\hat{e}$, amplitude $A$, frequency $f$, momentum $k$, and initial phase $\varphi$. These constitutive properties can also be the function of position $\vec{r}$ and time $t$, leading to a higher-dimensional structured wave. Wave science and technology development is intimately related to flexibly controlling and fully

utilizing these fundamental properties. Conventional EM wave manipulations primarily rely on the accumulated propagation effect in naturally existing dielectric materials such as lenses, optical modulators, and waveplates[8,9]. These optical components are bulky and generally have curved shapes, making them unsuitable for modern integrated electronic and photonic systems.

The advance of metasurfaces[10,11], allowing wave-matter interactions within an ultrathin artificial surface, provides a paradigm shift for EM-wave manipulations[12–18]. Since then, the science and engineering communities have long sought-after a dream 'universal' metasurface that enables simultaneous and independent controls over all the

[1]State Key Laboratory of Terahertz and Millimeter Waves, City University of Hong Kong, Hong Kong 999077, China. [2]Department of Electrical Engineering, City University of Hong Kong, Hong Kong 999077, China. [3]State Key Laboratory of Millimeter Waves, Southeast University, Nanjing 210096, China. [4]Institute of Electromagnetic Space, Southeast University, Nanjing 210096, China. [5]Frontiers Science Center for Mobile Information Communication and Security, Southeast University, Nanjing 210096, China. [6]Guangdong-Hong Kong Joint Laboratory for Big Data Imaging and Communication, Shenzhen 518048, China. [7]These authors contributed equally: Geng-Bo Wu, Jun Yan Dai. ✉e-mail: junyand@seu.edu.cn; qiangcheng@seu.edu.cn; tjcui@seu.edu.cn; eechic@cityu.edu.hk

fundamental properties of EM waves. Although some effects have been made (Table 1), a single metasurface component that can manipulate all EM waves' properties remains elusive due to the following challenges. Firstly, most metasurfaces are passive, whose functionalities are set in stone and cannot be altered post-fabrication. However, many modern wave-empowered applications such as communications, holographic displays, and light detection and ranging demand dynamic and active controls for environment adapting and/or information processing. Some tunable metasurfaces integrated with active functional materials have been explored to achieve dynamic wave controls upon the external stimuli, including the electrical bias[19–21], mechanical deformation[22–24], optical pumping[25–28], and thermal excitation[29–31]. However, most of the tunable metasurfaces hitherto can control only one or two wave properties due to insufficient degrees of freedom in the element's geometrical parameters and external control variables to support the regulation of all wave properties[32]. Another grand challenge is the difficulty of independent wave property manipulations, as the controls over these properties are generally coupled with each other. For instance, unique geometric structures and co-optimization with active materials should be adopted to decouple the amplitude and phase regulations[33–35], but the design

complexity and insertion loss are amplified geometrically with the increase of the control degrees of freedom.

In recent years, spatiotemporally modulated metasurfaces have become an emerging technique to engineer EM waves in both space and time. Spatiotemporally modulated metasurfaces have introduced an additional dimension, time, into the conventional metasurface design, enabling intriguing physical phenomena[36–41] and wave manipulations in frequency-momentum spaces[42–50]. In particular, the waveguide-integrated metasurface antenna allows high-efficiency frequency controls without sideband pollution[44,51,52]. However, only the frequency and momentum controls are validated for most spatiotemporally modulated metasurfaces (Table 1). Whether it is possible to control all of the fundamental properties of EM waves with a single radiation aperture remains an outstanding question. The situation is far more challenging to realize simultaneously and independently programmable radiation characteristics.

In this article, we report a microwave universal metasurface antenna (UMA) capable of dynamically, simultaneously, independently, and precisely manipulating all fundamental properties of EM waves, including amplitude, phase, polarization, frequency, and momentum. This is enabled by the additional degree of freedom

**Table 1 | Comparison of state-of-the-art metasurfaces in the literature**

| Ref. | Property | | | | | Wave mani. experimentation | | | | | Info. mani. experimentation | | |
|---|---|---|---|---|---|---|---|---|---|---|---|---|---|
| | Freq./wavelength | Tunable | Method | Type | SB free | Amp. | Phase | Mom. | Pol. | Freq. | MWG | Security | IDM |
| 10 | 8 µm | × | / | 3-bit | / | × | × | ✓ | × | × | × | × | × |
| 11 | 1.0–1.9 µm | × | / | 3-bit | / | × | × | ✓ | × | × | × | × | × |
| 15 | 750 THz | × | / | 1-bit | / | × | × | ✓ | × | × | × | × | × |
| 16 | 8.6 GHz | ✓ | Elect. bias | 1-bit | / | × | × | ✓ | × | × | × | × | × |
| 17 | 620 nm | ✓ | Elect. bias | 1-bit | / | × | × | ✓ | × | × | × | × | × |
| 19 | 8.5 µm | ✓ | Elect. bias | Cont. | / | × | ✓ | ✓ | × | × | × | × | × |
| 20 | 6 µm | ✓ | Elect. bias | Cont. | / | × | ✓ | × | ✓ | × | × | × | × |
| 21 | 0.3 THz | ✓ | Elect. bias | 3-bit | / | ✓ | ✓ | ✓ | × | × | × | × | × |
| 22 | 650 nm | ✓ | Mech. defo. | Cont. | / | ✓ | × | × | × | × | × | × | × |
| 23 | 632.8 nm | ✓ | Mech. defo. | Cont. | / | × | × | ✓ | × | × | × | × | × |
| 24 | 915 nm | ✓ | Mech. defo. | Cont. | / | × | × | ✓ | × | × | × | × | × |
| 25 | 800 nm | ✓ | Opt. pumping | 1-bit | / | ✓ | × | × | × | × | × | × | × |
| 26 | 860 nm | ✓ | Opt. pumping[STM] | Cont. | × | × | × | ✓ | × | ✓ | × | × | × |
| 29 | 1.64 µm | ✓ | Therm. excit. | Cont. | / | ✓ | × | × | × | × | × | × | × |
| 30 | 4.79 µm | ✓ | Therm. excit. | Cont. | / | × | ✓ | × | × | × | × | × | × |
| 31 | 0.75 THz | ✓ | Therm. excit. | Cont. | / | ✓ | × | × | × | × | × | × | × |
| 33 | 1550 nm | ✓ | Elect. bias | Cont. | / | × | ✓ | × | × | × | × | × | × |
| 34 | 7 µm | ✓ | Elect. bias | Cont. | / | ✓ | ✓ | × | × | × | × | × | × |
| 35 | 1.3 µm | ✓ | Elect. bias | Cont. | / | ✓ | ✓ | ✓ | × | × | × | × | × |
| 37 | 6.9 GHz | ✓ | Elect. bias[STM] | Cont. | × | × | × | ✓ | × | ✓ | × | × | × |
| 42 | 10 GHz | ✓ | Elect. bias[STM] | 1-bit | × | × | × | ✓ | × | ✓ | × | × | × |
| 43 | 9.5 GHz | ✓ | Elect. bias[STM] | 2-bit | × | ✓ | × | ✓ | × | ✓ | ✓ | ✓ | × |
| 44 | 27 GHz | ✓ | Elect. bias[STM] | 1-bit | ✓ | × | × | ✓ | × | ✓ | × | × | × |
| 45 | 3.7 GHz | ✓ | Elect. bias[STM] | Cont. | × | ✓ | × | ✓ | × | ✓ | × | × | × |
| 46 | 4 GHz | ✓ | Elect. bias[STM] | Cont. | × | × | ✓ | × | × | ✓ | ✓ | × | × |
| 47 | 4.25 GHz | ✓ | Elect. bias[STM] | Cont. | ✓ | × | ✓ | ✓ | × | ✓ | × | × | × |
| 48 | 2.6 GHz | ✓ | Elect. bias[STM] | Cont. | ✓ | × | ✓ | ✓ | × | ✓ | × | × | × |
| 49 | 2.6 GHz | ✓ | Elect. bias[STM] | Cont. | × | ✓ | ✓ | × | ✓ | ✓ | × | × | × |
| 50 | 2.7 GHz | ✓ | Elect. bias[STM] | Cont. | × | × | ✓ | ✓ | × | ✓ | ✓ | × | × |
| 56 | 2.15 GHz 5.34 GHz | ✓ | Elect. bias[STM] | 1-bit | × | ✓ | × | ✓ | × | × | ✓ | ✓ | × |
| This work | 23.5 GHz | ✓ | Elect. bias[STM] | 1-bit | ✓ | ✓ | ✓ | ✓ | ✓ | ✓ | ✓ | ✓ | ✓ |

*SB* sideband, *STM* spatiotemporal modulation, *MWG* modulated waveform generation, *IDM* inherent direction modulation.

provided by the spatiotemporal modulation and the unique waveguide-integrated structure of the UMA. We show that our UMA facilitates space-varying wave properties for more complicated wave manipulation by demonstrating the Airy beam and focusing beam generation. Our UMA can also enable time-varying wave properties for information manipulation. For validation, we build a wireless communications link in which our UMA directly transmits the modulated waveforms with different modulation schemes. We further reveal our UMA's unique direction information modulation phenomenon, making it an ideal candidate for eavesdropper-proof communications. The complicated wave manipulations and information modulations are achieved via spatiotemporally switching the meta-atoms' ON-OFF (1-bit) coding states. Our UMA offers a universal EM platform for wave and information manipulations, making it a promising enabler for next-generation wireless communications, integrated photonics, and quantum information science.

## Results

### Full manipulations of all fundamental EM-wave properties

Our UMA consists of an array of subwavelength anisotropic meta-atoms on top of a waveguiding structure (Fig. 1a). In this work, we consider the metasurface antenna operating at microwave frequencies and utilize the positive-intrinsic-negative (PIN) diode and substrate-integrated waveguide (SIW) as the active element and waveguide, respectively (Fig. 1b). Each meta-atom consists of two $\pm45^\circ$-inclined slot openings to radiate two orthogonal eigen-polarization states $|u\rangle$ ($+45^\circ$ linear polarization) and $|v\rangle$ ($-45^\circ$ linear polarization) in free space. Each slot opening can be independently switched between the radiating ('1') and non-radiating ('0') states in real time by the PIN diode. Assume that the radiating state of the meta-atom is temporally modulated with a time cycle $T_M = 1/f_M$, and the frequency of the injected monochromatic wave is $f_0$, subject to $f_0 \gg f_M$. Applying independent time-coding sequences to all meta-atoms forms a two-dimensional (2D) "0/1" space-time-coding (STC) matrix[42–44] (see Supplementary Fig. S1b), which is controlled by a field programmable gate array (FPGA). Our STC metasurface antenna enables extracting and converting the in-plane guided wave (GW) into the out-of-plane propagating wave (PW) with arbitrary wave properties.

We first demonstrate the independent controllability of our UMA over all of the fundamental wave properties. Frequency manipulation is generally challenging due to the requirement to change a photon's energy. The typical approach based on nonlinear bulk media suffers from the weak nonlinear effect and stringent phase-matching conditions. Here, the functions of our STC meta-atom are twofold: (i) temporal modulation for producing nonlinear effects; and (ii) simultaneous space-time modulations for converting the newly generated waves into free space to mitigate the phase matching condition (Fig. 1a). Specifically, the periodic switching of the meta-atoms' radiation states leads to an infinite number of harmonic frequencies $f_0 + mf_M$, where $m$ is an arbitrary integer. We apply an identical rectangular time sequence yet with a position-dependent time shift[42,45] $t_i(x)$ to the meta-atoms (Supplementary Fig. S1c). According to the time-shifting property of the Fourier Transform, a shift in the time domain $t_i$ corresponds to a linear phase shift $-2\pi mf_M t_i$ in the frequency domain. Therefore, the total phase shift of the radiated PW consists of two parts (see Supplementary Fig. S1d): (i) the phase accumulation from the propagation of GW $\varphi_{GW} = -\xi_{GW}x_i$ (where $\xi_{GW}$ is the wavenumber inside the waveguide); and (ii) the abrupt phase shift induced by the spatio-temporal modulation $\varphi_{ST} = -2\pi mf_M t_i(x)$. The corresponding linear momentum of the radiated PW along the $x$ direction is $k_x(f_0 + mf_M) = \xi_{GW} + k_{ST}$, where $k_{ST} = 2\pi mf_M \partial t_i(x)/\partial x$ is the additional momentum imparted by the spatiotemporal modulation (Fig. 1a). In this illustrative example (Fig. 1a), we leverage this momentum to compensate for the momentum mismatch between the waveguide and free space at the target $m = +1$ harmonic frequency, i.e., $-\xi_{m=+1} < \xi_{GW} + k_{ST} < \xi_{m=+1}$,

where $\xi_{m=+1}$ is the free-space wavenumber at the $m = +1$ harmonic frequency. Other unwanted harmonics are not supported and are highly suppressed in both free space and the waveguide due to the tremendous momentum mismatch (see Supplementary Fig. S2 for the measured spectrum distribution). In this manner, our UMA can simultaneously achieve a nearly perfect GW-to-PW conversion and frequency shifting. Figure 1c presents the extracted free-space wave's measured spectra with different frequency values shifting from –1.8 MHz to +1.8 MHz through changing the modulation frequency $f_M$ by the FPGA.

Next, we show the momentum control of the up-converted PW (Fig. 1d). The linear momentum imparted by the spatiotemporal modulation $k_{ST} = 2\pi mf_M \partial t_i(x)/\partial x$ is proportional to the applied time gradient $\partial t_i(x)/\partial x$. Supposing $\partial t_i(x)/\partial x$ is a constant, the radiated PW has a well-defined radiation angle $\theta_r = \sin^{-1}(k_x/\xi_m)$ (see Methods). Therefore, the momentum and the corresponding output angle of the radiated PW can be easily tuned by changing the applied time gradient[42] (Fig. 1d). Moreover, since the abrupt phase shift imparted by the spatiotemporal modulation satisfies $\varphi_{ST} = -2\pi mt_i/T_M$, the initial phase of the extracted PW can be tuned from $0^\circ$ to $360^\circ$ by altering the reference time shift $t_{i=1}$ (the time shift for the 1st meta-atom) while fixing the time gradient $\partial t_i(x)/\partial x$ (Fig. 1e). Furthermore, the harmonic frequencies' power distribution depends on the time sequence's coding context (see Methods). The amplitude of the extracted PW can be tuned by varying the duty cycle $\tau$ of the rectangular time sequence for all meta-atoms (Fig. 1f). Finally, it is known that arbitrary polarizations can be decomposed into a linear combination of two complete orthogonal polarization bases, e.g., $|u\rangle$ and $|v\rangle$, and vice versa. The anisotropic meta-atom consists of a pair of $\pm45^\circ$-inclined elliptical slot openings with a large length-to-width ratio, whose radiated electric field polarization is perpendicular to the long side (Fig. 1b and Supplementary Fig. S1a). Applying independent STC matrixes to the $\pm45^\circ$-inclined slots in each meta-atom allows independent control over the amplitude and phase contents of the extracted $|u\rangle$ and $|v\rangle$ components, resulting in arbitrary polarization generation by our UMA. Specifically, an identical time gradient $\partial t_i(x)/\partial x$ is adopted for both the $\pm45^\circ$-inclined slot openings such that the $|u\rangle$ and $|v\rangle$ components share the same momentum and output direction. We apply different initial time delays and duty cycles to the $\pm45^\circ$-inclined slot openings to control the amplitude ratio and phase difference between the radiated $|u\rangle$ and $|v\rangle$ components, respectively. Six representative polarizations, including $|x\rangle$, $|y\rangle$, $|u\rangle$, $|v\rangle$, $|LCP\rangle$ (left-hand circular polarization), and $|RCP\rangle$ (right-hand circular polarization) are given as illustrative examples to show the polarization controllability (see Fig. 1g and Supplementary Table S1). The coupling effects between the two $\pm45^\circ$-inclined slot openings for polarization control are given in Supplementary Note 5. The required STC matrixes for different polarizations are given in Supplementary Fig. S9a.

Our UMA can also independently manipulate all the properties of the radiated PW without shifting the frequency (Fig. 1h–l and Methods). We leverage the time-average effect of the spatiotemporal modulation to form an equivalent sinusoidal amplitude distribution at the fundamental frequency (see Methods). In this manner, the $n = -1$ space harmonic becomes a fast wave and converts to a PW in free space (Fig. 1h). The momentum and the corresponding output angle of the extracted PW can be tuned by varying the spatial period $\Lambda$ through changing the applied STC matrix (Fig. 1i). Next, we utilize the space-shifting property of the Fourier Transform to control the phase of the extracted PW. In analogy with the time-shifting property but in the space domain, a shift in the space domain $\Delta x$ results in a phase shift $2\pi n \Delta x/\Lambda$ for the $n$th-order space harmonic. We apply different space shifts $\Delta x$ to the sinusoidal amplitude envelope, resulting in different phases to the radiated $n = -1$ space harmonic (Fig. 1j). The extracted power for each meta-atom from the waveguide is proportional to the modulation efficiency of the sinusoidal amplitude modulation

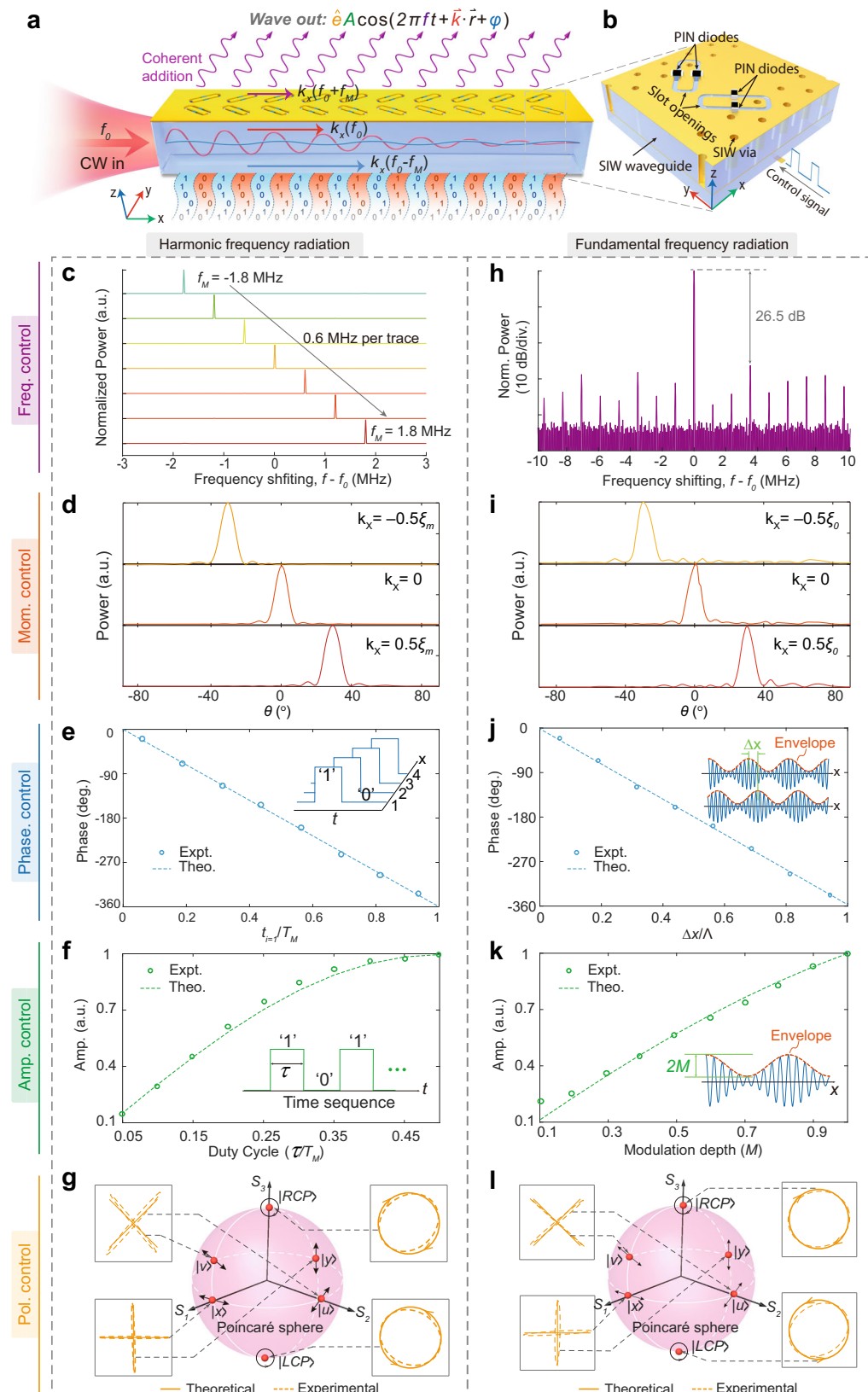

$\eta = \frac{\left(\frac{M}{2}\right)^2}{1 + 2\left(\frac{M}{2}\right)^2} = \frac{M^2}{4 + 2M^2}$. Therefore, the amplitude of the extracted PW of the UMA can be tuned by changing the modulation depth $M$ (Fig. 1k). Finally, our UMA can generate arbitrary polarization by controlling the amplitude ratio and phase difference of the extracted $|u\rangle$ and $|v\rangle$ components. To this end, we apply equivalent sinusoidal amplitude

distributions with an identical spatial period $\Lambda$ to both the ±45°-inclined slot openings such that the extracted $|u\rangle$ and $|v\rangle$ components share the same output angle in free space. We apply different space translations and modulation depths to the ±45°-inclined slot openings to control the amplitude ratio and phase difference of the $|u\rangle$ and $|v\rangle$ components, respectively. As proof-of-concept examples, the UMA is

**Fig. 1 | The UMA for independent controls of all fundamental wave properties.**
**a** Schematic of the UMA. Slot-opening meta-atoms are positioned on top of the waveguide to convert the GW into PW with software-defined properties. By applying 1-bit '0/1' space-time-coding sequences to switch the meta-atom between the radiating ('1') and non-radiating ('0') states, we can control all fundamental properties of the radiated wave. Only the momentum of the target harmonic frequency ($m = +1$ harmonic in this case) matches with that of free space. In contrast, other unwanted higher-order harmonics are highly suppressed in both the waveguide and free space without phase-matching. **b** Configuration of the anisotropic meta-atom, consisting of two $\pm 45°$-inclined slot openings loaded with PIN diodes. **c-g** Independent control of wave properties for the $m = +1$ harmonic frequency radiation. **c** Measured frequency shifting of the extracted free-space PW with different modulation frequencies $f_M$. The momentum control (**d**), phase control (**e**),

amplitude control (**f**), and polarization control (**g**) are achieved by tuning the time gradient $\partial t_i(x)/\partial x$, reference time shift $t_{i=1}$, duty cycle $\tau$, and time gradient and reference time shift for the $|u\rangle$, $|v\rangle$ polarizations, respectively. **h-l** Independent control of wave properties for the $m = 0$ fundamental frequency radiation. **h** Measured spectrum of the radiated PW for $m = 0$ fundamental frequency radiation. The momentum control (**i**), phase control (**j**), amplitude control (**k**), and polarization control (**l**) are achieved by tuning the equivalent spatial period $\Lambda$, space shift $\Delta x$ of the amplitude envelope, modulation depth $M$, and spatial period and space shift for the $|u\rangle$, $|v\rangle$ polarizations, respectively. The input frequency and modulation frequency in (**d-l**) are $f_0 = 23.5$ GHz and $f_M = 1.2$ MHz, respectively. Here, CW continuous wave, SIW substrate-integrated waveguide, PIN positive-intrinsic-negative.

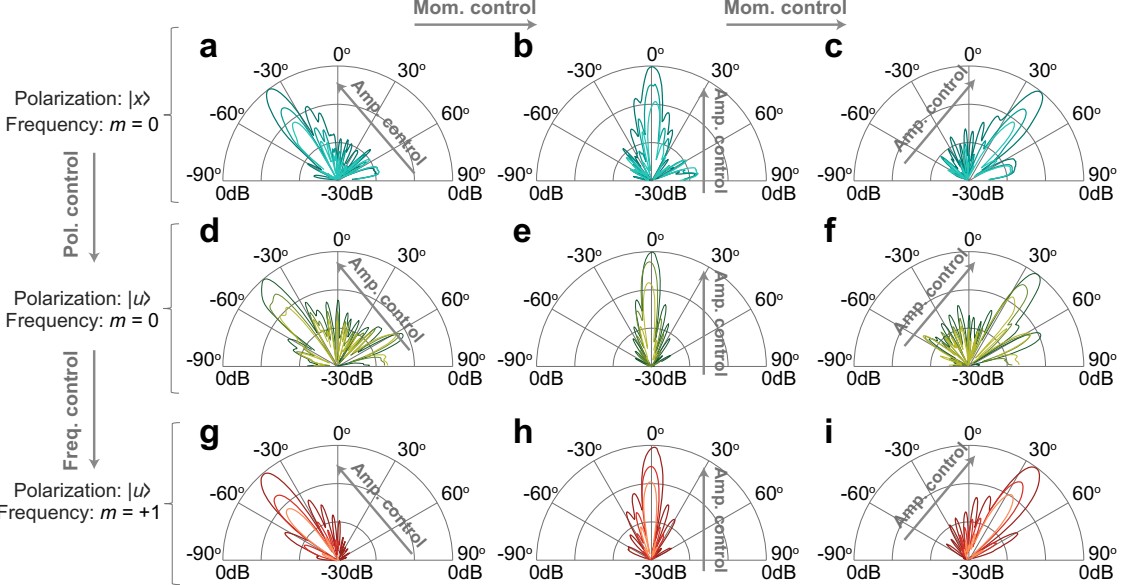

**Fig. 2 | The UMA for simultaneous manipulations of the radiated EM wave properties. a-c** Measured far-filed radiation patterns of the UMA at the fundamental frequency with the $|x\rangle$ polarization, whose main beam scans to $-40°$ (**a**), $0°$ (**b**) and $40°$ (**c**), respectively. **d-f** Same as (**a-c**), except that the polarization is

changed to $|u\rangle$. **g-i** Same as (**d-f**), except that the frequency of the extracted PW is changed to the $m = +1$ harmonic. Three curves in each panel demonstrate the amplitude controls of the radiated PWs. The input frequency and modulation frequency are $f_0 = 23.5$ GHz and $f_M = 1.2$ MHz, respectively.

loaded with different STC matrixes (see Supplementary Fig. S9b) to generate six representative polarizations, including $|x\rangle$, $|y\rangle$, $|u\rangle$, $|v\rangle$, $|LCP\rangle$, and $|RCP\rangle$ (Fig. 1l and Supplementary Table S1).

We further show the *simultaneous* controls over EM wave's properties by investigating the far-field radiation patterns of the UMA. In each panel in Fig. 2, three radiating cases with different magnitudes are given to demonstrate the independent amplitude controllability. The panels in the same row in Fig. 2 represent the radiated PWs with identical wave properties except with different linear momentums, whose corresponding beam angle steers from $-40°$, $0°$ to $+40°$, verifying the simultaneous amplitude and momentum controls. We change the polarization from $|x\rangle$ in the first-row panels to $|u\rangle$ in the second-row panels, leading to simultaneous amplitude, momentum, and polarization controls. We further change the output frequency from the fundamental frequency ($m = 0$) in the 2nd-row panels to $m = +1$ harmonic frequency in the 3rd-row panels while fixing all other wave properties. In this case, simultaneous amplitude, momentum, polarization, and frequency controls have been verified.

## Space-varying wave properties for complicated wave manipulation

We have demonstrated the full-dimensional tailoring of the EM waves yet with spatial-invariant (uniform) wave properties. Here, we show that our UMA can generate more complicated EM waves with space-varying

wave properties for more complicated waveform generation and beamforming. As the first illustrative example, our metasurface antenna can generate an Airy beam with phase-varying properties (Fig. 3a). The required phase profile along the metasurface aperture should fulfill the following equation to generate a parabolic trajectory $x \propto z^2$ [53]

$$\varphi(x) = \frac{4}{3} a^{\frac{1}{2}} \xi_m (-x)^{3/2} \tag{1}$$

where $a$ is the acceleration factor and $\xi_m$ is the free-space wavenumber at the target $m$th-order harmonic frequency. For the UMA, the phase distribution of the extracted wave at the $m$th-order harmonic frequency is the sum of the accumulated phase shift of the guided wave and the equivalent phase shift imparted by the spatiotemporal modulation, given by $\varphi_{GW} + \varphi_{ST} = -\xi_{gw}x - 2\pi m f_M t_i(x)$ (Supplementary Fig. S1d). Combining with Eq. (1), the required normalized time shift of the UMA is given by

$$\frac{t_i(x)}{T_M} = \frac{\frac{4}{3} a^{\frac{1}{2}}_{\xi_m} (-x)^{3/2} + \xi_{gw} x}{-2\pi m} \tag{2}$$

The calculated time shift for generating the Airy beam with the acceleration factor $a = 0.003$ at the $m = -1$ harmonic frequency is given

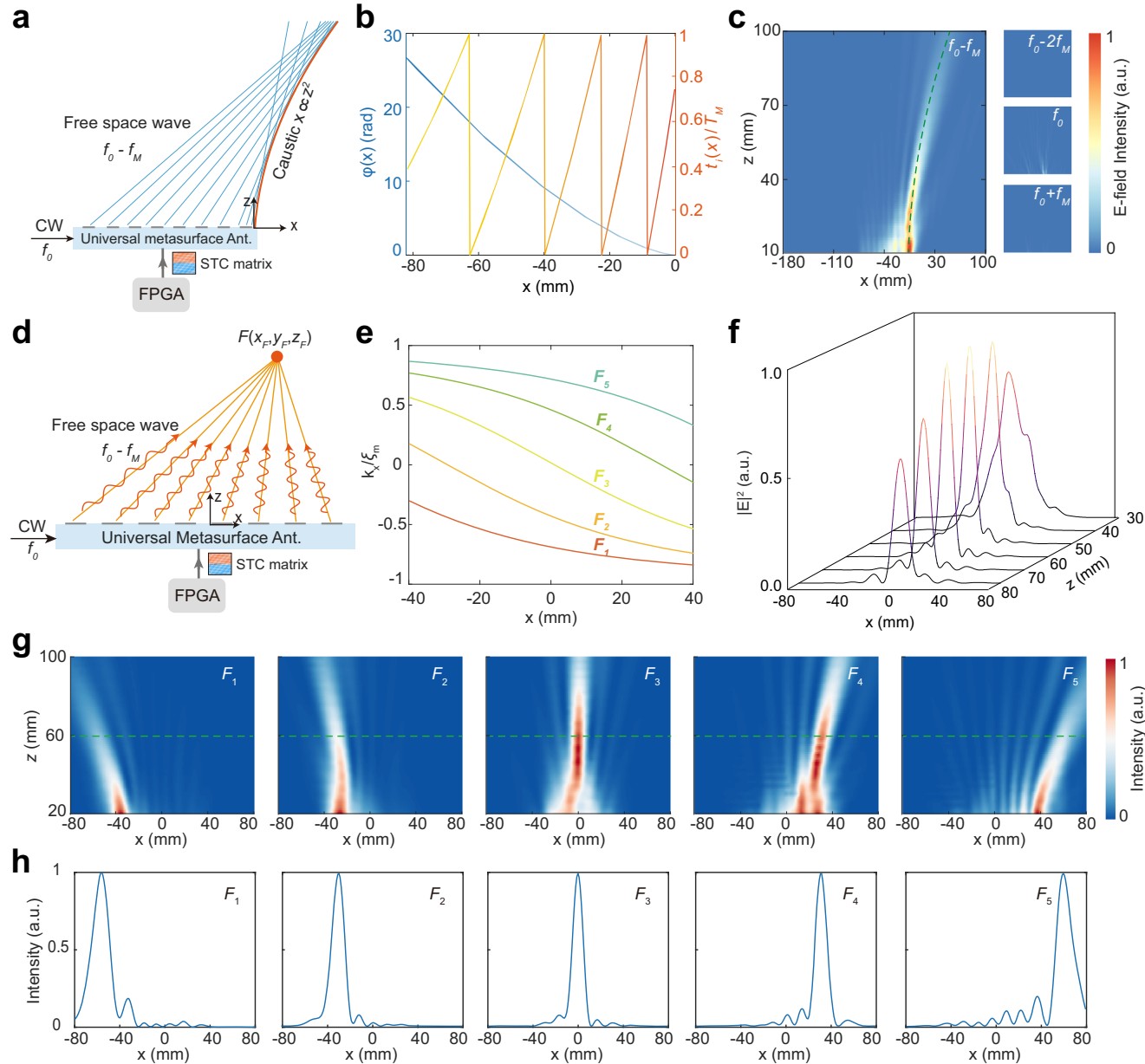

**Fig. 3 | The UMA for complicated beam shaping with space-varying wave properties. a**–**c** UAM for Airy beam generation. **a** Schematic illustration of the UMA for generating the Airy beam at the $m = -1$ harmonic frequency in free space. The Airy beam requires a spatial-varying phase profile as $\frac{4}{3}a^{1/2}\xi_m(-x)^{3/2}$. **b** The required aperture phase profile of the Airy wavefront and the corresponding normalized time shifting for the meta-atoms at different positions. **c** The measured E-field intensities in the $xz$-plane at different harmonic frequencies. The green dashed line represents the theoretical parabolic trajectory of the Airy beam $x = az^2$, where the acceleration factor $a = 0.003$ in this case. Only the target $m = -1$ harmonic frequency possesses high field intensity, whereas other harmonic frequencies are all highly suppressed in free space. **d**–**h** UAM for focused beam generation. **d** Schematic illustration of the UMA for wave focusing at the $m = -1$ harmonic frequency. In this case, the local momentum or output angle of the extracted PWs from all meta-atoms are different. **e** The required normalized local momentum $k_x/\xi_m$ as a function of the meta-atom position for different focal spot positions from $F_1$ to $F_5$, where $F_1 = (-60, 0, 60)$ mm, $F_2 = (-30, 0, 60)$ mm, $F_3 = (0, 0, 60)$ mm, $F_4 = (30, 0, 60)$ mm, and $F_5 = (60, 0, 60)$ mm. **f** Measured cross-section intensity distributions at the $m = -1$ harmonic frequency in different transversal planes for the designed focal point $F_3$ (0, 0, 60) mm. **g** The measured E-field intensities in the $xz$-plane at the $m = -1$ harmonic frequency for different focus spots. **h** The corresponding 1-D normalized intensity distributions on the focal plane $z = 60$ mm. The input frequency and modulation frequency are $f_0 = 23.5$ GHz and $f_M = 1.2$ MHz, respectively.

in Fig. 3b. Figure 3c presents the measured electric field intensity distributions at different harmonic frequencies. We observe that the UMA can extract and mould the waves into an accelerating and non-diffractive Airy beam with a well-defined parabolic trajectory. Moreover, other undesired harmonic frequencies are highly suppressed in free space.

Our metasurface antenna can also mould the extracted waves with space-varying momentum properties $k_x(x)$ for wave-focusing

applications (Fig. 3d). To focus the extracted wave into a desired focal point $F = (x_F, z_F)$, the linear momentum of the extracted PW should fulfill the following formula based on the geometric ray and the geometrical relationship (Supplementary Fig. S4a)

$$k_x(x) = \xi_m \frac{x_F - x}{\sqrt{(x_F - x)^2 + (z_F - z)^2}} \tag{3}$$

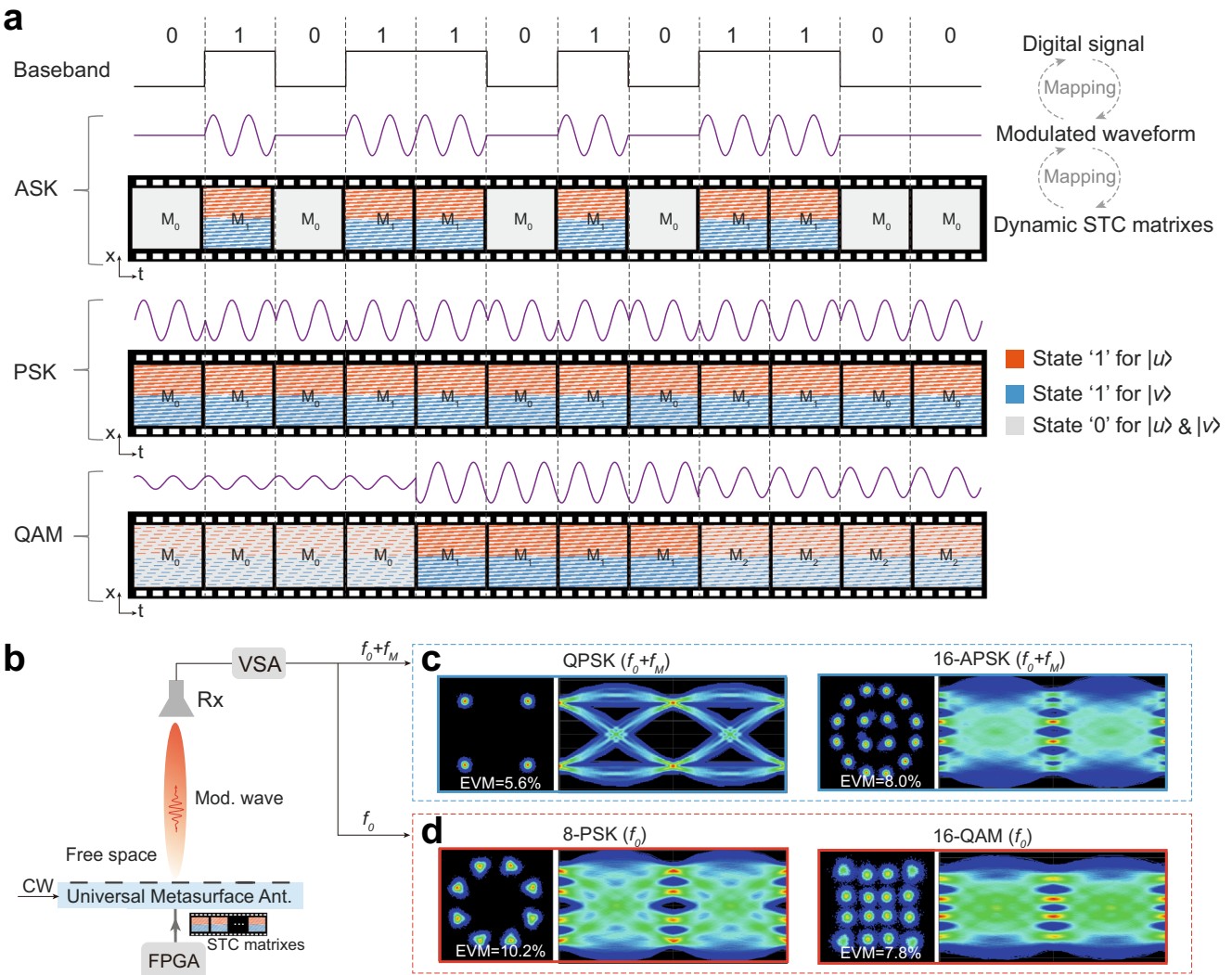

**Fig. 4 | The UMA for information modulation with time-varying wave properties. a** One-to-one mapping relationship among the transmitted digital information stream, the radiated information-carried waveforms in the free space (2ASK, BPSK and 16QAM), and the applied dynamic STC matrixes for communications at the $m = +1$ harmonic frequency. The required dynamic STC matrixes for generating modulated waveform at the fundamental frequency are presented in Supplementary Fig. S5b. **b** The UMA is used to directly generate the information-carried EM waveforms in free space. **c, d** The measured decoded constellation diagrams and eye diagrams at the receiver end for communications at the $m = +1$ (**c**) and $m = 0$ (**d**) frequencies, respectively. CW continuous wave, Mod modulated, Ant. antenna, VSA vector signal analyzer, EVM error vector magnitude.

The corresponding required time gradient to achieve such momentum of the extracted PW is

$$\partial t_i(x)/\partial x = \left[ \xi_m \frac{x_F - x}{\sqrt{(x_F - x)^2 + (z_F - z)^2}} - \xi_{gw} \right] / (2\pi m f_M) \quad (4)$$

As illustrative examples, we consider the UMA generates different intended focal points from $F_1$ to $F_5$ at the $m = -1$ harmonic frequency, whose required spatial-varying linear momentums are presented in Fig. 3e. The corresponding 2-D field intensities distributions at the $m = -1$ harmonic frequency are presented in Fig. 3g, respectively. We observe that the metasurface antenna can extract and mould the PW wave into intended focal spots, whose positions are software-defined according to the applied STC matrix. Figure 3f shows the measured cross-section intensity distribution in different transversal planes for the designed focal point $F_3$ (0, 0, 60) mm. We can observe that the E-field intensity increases as close to the focal plane. Due to the free-space wave spreading effect, the density peak is located at around

$z = 50$ mm, which is between the designed focal point and antenna's aperture[54]. Nevertheless, the strongest field on the interested focal plane $z = 60$ mm is exactly located at the designed focal position for all the focusing scenarios (Fig. 3h), making it suitable for microwave real-time imaging systems[55]. The effects of the 1D and 2D radiating aperture on the fading distance for the Airy beam and focusing beam are investigated in Supplementary Note 6. The wave focusing effect of the UMA is weaker as the designed focus moves away from the radiating aperture, as shown in Supplementary Fig. S15. Our UMA's flexible and agile beam-shaping capability holds promising potential in sensing, imaging, and wireless power transfer applications.

**Time-varying wave properties for information manipulation**
We further demonstrate that our UMA enables information manipulation by generating time-varying wave properties. Figure 4a presents the digital baseband information and the corresponding transmitted modulated PWs in free space for some popular modulation schemes in modern wireless communications systems. The amplitude-shift keying (ASK), phase-shift keying (PSK), and quadrature amplitude modulation

(QAM) schemes map to the free-space PWs with time-varying amplitude, time-varying phase, and time-varying amplitude and phase properties, respectively. Generation of such modulated waveforms in free space by conventional radiofrequency transmitters relies on a heterodyne architecture consisting of a series of active/passive modules, including digital to analog converters, modulators, mixers, filters, phase shifters, and antenna arrays to mould the PW with time-varying amplitude and phase (information-carried) properties. In contrast, our UMA−a single component only− can directly generate identical modulated waveforms with time-varying amplitude and phase properties in a single step. This can be achieved by loading the dynamic STC matrices, mapping to the desired digital information stream to be transmitted (Fig. 4a). This provides a radically new communication paradigm in the physical layer with the advantages of a much simpler structure, higher integration, lower cost, and lower power consumption. Figure 4c, d presents the measured decoded constellation diagrams and eye diagrams at the receiver end when the information is carried at the $m = +1$ and $m = 0$ harmonic frequencies, respectively. The information manipulation functionality of our UMA is validated for different information modulation schemes, including QPSK, 8PSK, 16 amplitude and phase-shift keying (16APSK), and 16QAM with transmission data rates of 1 Mbps, 1.5 Mbps, 2 Mbps, and 2 Mbps, respectively.

Moreover, the control over other properties of EM waves (momentum, frequency, and polarization) by our UMA opens additional opportunities to achieve space-division multiplexing (SDM), frequency-division multiplexing (FDM), and polarization-division multiplexing (PDM), which can establish multiple independent channels to improve the communications capacity. We set up a joint SDM-FDM-PDM data transmission link (Figs. 5a and 6c), in which our UMA simultaneously transmits two independent channels with different beam directions (–30º and +30º for SDM), different polarizations ($|u\rangle$ and $|v\rangle$ for PDM), different frequencies ($m = -1$ and $m = -2$ harmonics for FDM), and different information modulation schemes (QPSK and 8PSK). To this end, we divide the UMA into two interwoven sub-metasurfaces (shown in Fig. 5a with two different colors). Each sub-metasurface, loaded with independent dynamic STC matrixes, generates one specific modulated waveform corresponding to one communication channel. Specifically, we apply two different time gradients to the two sub-metasurfaces; the equivalent momentums imparted by the spatiotemporal modulation push the $m = -1$ and $m = -2$ harmonic frequencies to radiate into the free space with output angles of –30º and 30º for the two sub-metasurfaces, respectively. Moreover, the amplitude and phase contents (carried information) of the two extracted PWs can be independently controlled by the applied duty cycle and reference time shift according to the transmitted binary bit streams of the two channels. The polarization states of the two radiated beams can be controlled by applying different space-time coding sequences to the ±45º-inclined slot openings in each sub-metasurface. Each sub-metasurface's bit stream is first mapped to the corresponding dynamic STC matrixes according to the desired conversion harmonic frequency, polarization, and signal modulation format. The final dynamic STC matrixes for dual-channel multiplexing is the sum of the two dynamic STC matrixes for the two interwoven sub-metasurfaces (see Supplementary Fig. S6). The lattice of meta-atoms in the sub-metasurface is $0.31\lambda_O$ (where $\lambda_O$ is the free-space wavelength at 23.5 GHz). The sub-metasurface is free of higher-order diffractions in the free space (Fig. 5b). Moreover, we can flexibly change the radiation directions of CH1 and CH2 (Fig. 5c). This can be achieved by changing the applied time gradients of the two sub-metasurfaces to vary the momentum property of the extracted PWs. The measured radiation patterns (Fig. 5b, c) and the measured decoded constellation diagrams of the two channels at the receiver end (right inset of Fig. 5b, c) verify that our UMA can realize simultaneous and independent multiplexing successfully.

In conventional transmitter architectures, the radiated EM waves in different directions hold identical time-varying wave properties (information). Therefore, an eavesdropper can recover the information even in the sidelobe region by using a sufficiently sensitive receiver (Supplementary Fig. S8c, d). Here, we reveal an intriguing inherent directional modulation (IDM) phenomenon of our UMA at the fundamental frequency. This unique IDM property is attributed to the direction-dependent phase control of our UMA based on spatial modulation (see Methods and Supplementary Fig. S7 for a detailed discussion). For validation, we establish a communication link, where our UMA directly emits the modulated waveforms carrying the 8PSK signals at the fundamental frequency (Fig. 5d). Figure 5e presents the corresponding measured radiation pattern and the decoded constellation diagrams when the receiver is located in different directions. We clearly note that only the receiver in the main-beam direction of the UMA can successfully decode the information, while the eavesdroppers outside the main beam lose the information entirely. This IDM effect can effectively mitigate malicious eavesdropper attacks in different directions, establishing physical-layer security on top of wireless communications. The IDM is an intrinsic property of our UMA and is free of any optimization and performance trade-off generally incurred for conventional approaches to achieve physically secured links[43,56].

## Discussion

In summary, we proposed and demonstrated a UMA capable of dynamically, simultaneously, independently, and precisely manipulating all of the fundamental properties of EM waves. We further demonstrated that our UMA could facilitate complicated wave and information manipulations. The UMA can directly generate the modulated waveforms with dynamic wave properties, leading to a paradigm shift for new information-transmitting architectures. The UMA concept can be extended to a 2-D aperture by periodically repeating the 1-D metasurface antenna along the $y$-axis, fed by a power-dividing network (See Supplementary Note 4 for the detailed metasurface configuration and 2D wavefront engineering).

Our UMA uniquely combines a number of significant merits, including full-dimensional wave controllability, inherent IDM, simplified coding scheme (1-bit), free of sideband pollution, and potential on-chip integration, making it an appealing enabler for the next-generation large-capacity and high-security information systems. Although demonstrated at the microwave band, the proposed concept can be extended to terahertz frequencies based on CMOS chips[21], monolayer molybdenum disulfide switches[57], and high-electron-mobility InAlN/GaN-based metadevices[58]. In the optical band, the nanoscale switch can be implemented by phase-changed materials, such as germanium antimony telluride[59,60]. These technologies greatly extend the applications of the UMA in augmented reality, holography, integrated sensing and communications for 6 G, quantum optics, and quantum information science.

## Methods
### Wave manipulations with frequency shifting
Here we extend the theoretical model of the waveguide-integrated metasurface antenna[44] to a more generalized one by taking the polarization into account. In contrast to the continuous field modeling in ref. 44, we describe the metasurface by an array of subwavelength scatterers with the discrete nature of the metasurface. The UMA consists of an array of slot-opening meta-atoms etched on the top conductive layer of the SIW waveguide along the $x$-axis (Supplementary Fig. S1a). The anisotropic slot-opening meta-atom is equivalent to two orthogonally orientated waveguide-fed magnetic dipoles to extract energy from the waveguide and radiate to free space. The wave properties of the extracted $|u\rangle$ and $|v\rangle$

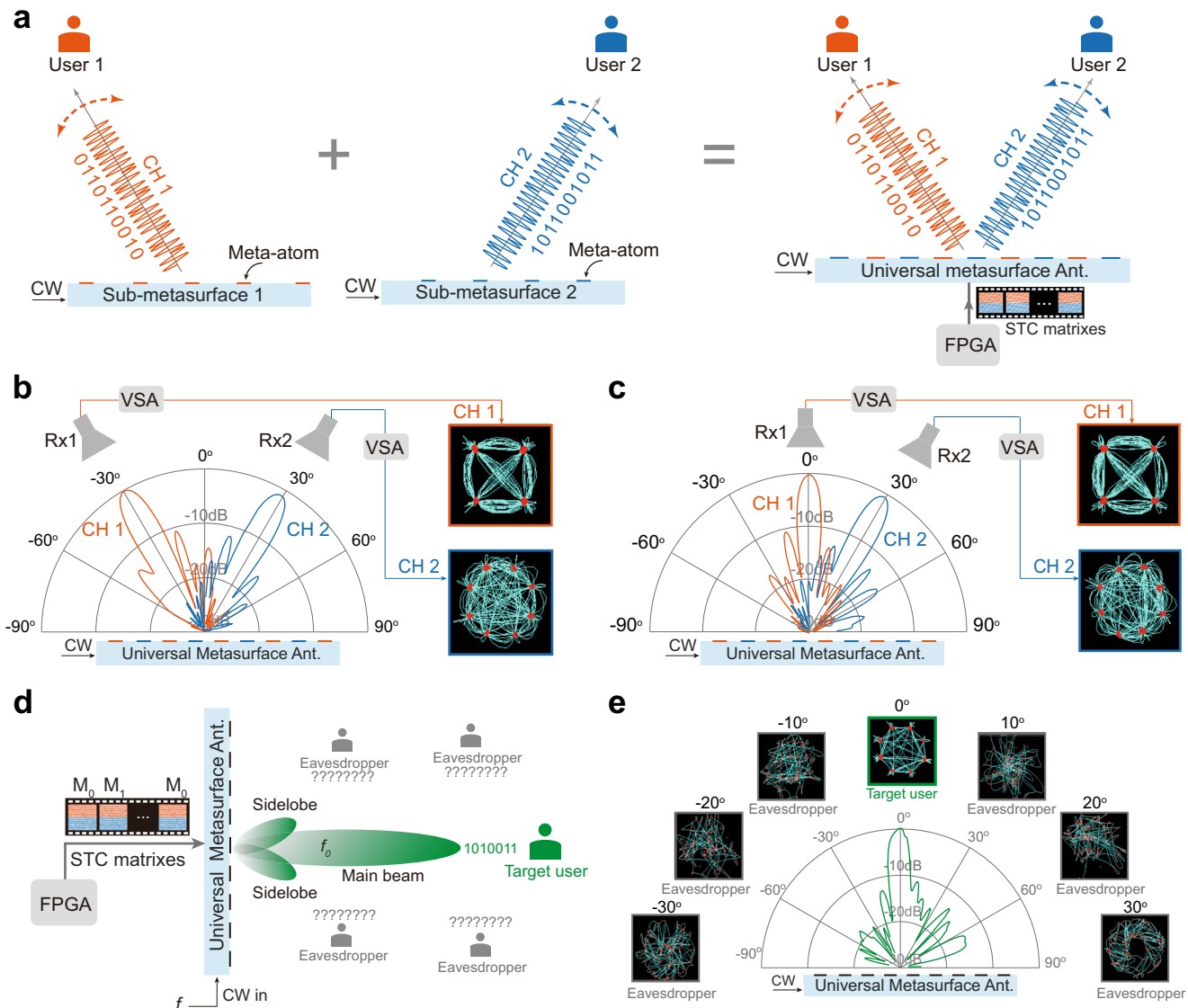

**Fig. 5 | The UMA for multi-channel and physical-layer security communications. a** The concept illustration of the UMA for directly generating two independently modulated waveforms carrying different digital data streams. Each sub-metasurface is responsible for one modulated waveform generation. Two designated users at different locations can simultaneously and independently receive the information from the UMA. **b, c** Measured radiation patterns of the UMA at the $m = -1$ (CH1) and $m = -2$ (CH2) harmonic frequencies. The insets on the right present the measured decoded constellation diagrams of the two channels at the receiver end. The beam directions for the CH1 and CH2 are $(-30°, 30°)$ and $(0°, 30°)$ in (**b**) and (**c**), respectively. **d** The concept illustration of the UMA with the inherent direction modulation property at the fundamental frequency. Only the target user in the main beam direction can successfully decode the information, whereas eavesdroppers at other positions totally lose the information. **e** Measured radiation pattern of the UMA at the fundamental frequency, and the measured decoded constellation diagrams when the receiver is located in different directions. The physical-layer security communication link for other main-beam directions is validated in Supplementary Fig. S8a, b. CW continuous wave, Mod modulated, Ant. antenna, VSA vector signal analyzer, CH channel, EVM error vector magnitude.

components can be independently controlled by applying independent control voltages to the PIN diodes incorporated in the meta-atom (Supplementary Fig. S1b). The excited polarized magnetic dipole moments $\bar{m}_i = \begin{bmatrix} m_i^u \\ m_i^v \end{bmatrix}$ with $|u\rangle$ and $|v\rangle$ components for the *i*th meta-atom at the position $x_i$ is

$$\bar{m}_i = \bar{\bar{P}}_i H_i = \bar{\bar{P}}_i(t) H_0 e^{-j\xi_{gw}x_i} e^{j2\pi f_0 t} \tag{5}$$

where $\bar{\bar{P}}_i = \begin{bmatrix} P_i^u(t) \\ P_i^v(t) \end{bmatrix}$ is the magnetic polarizability Jones matrix at instant $t$, and $H_i$ is the magnetic field of the reference guided wave inside the waveguide. Since the radiating state of the meta-atom is periodically

ON-OFF switched with a time cycle $T_M$, the magnetic polarizability is a periodic function of time, satisfying $\bar{\bar{P}}_i(t) = \bar{\bar{P}}_i(t + T_M)$, which can be decomposed into a Fourier series

$$\bar{\bar{P}}_i(t) = \sum_{m=-\infty}^{\infty} \bar{\bar{p}}_{i,m} e^{j2\pi m f_M t} \tag{6}$$

where $f_M = 1/T_M$ is the modulation frequency. The Fourier coefficients $\bar{\bar{p}}_{i,m} = \begin{bmatrix} p_{i,m}^u \\ p_{i,m}^v \end{bmatrix}$ can be calculated by

$$\bar{\bar{p}}_{i,m} = \frac{1}{T_M} \int_0^{T_M} \bar{\bar{P}}_i(t) e^{-j2\pi m f_M t} dt \tag{7}$$

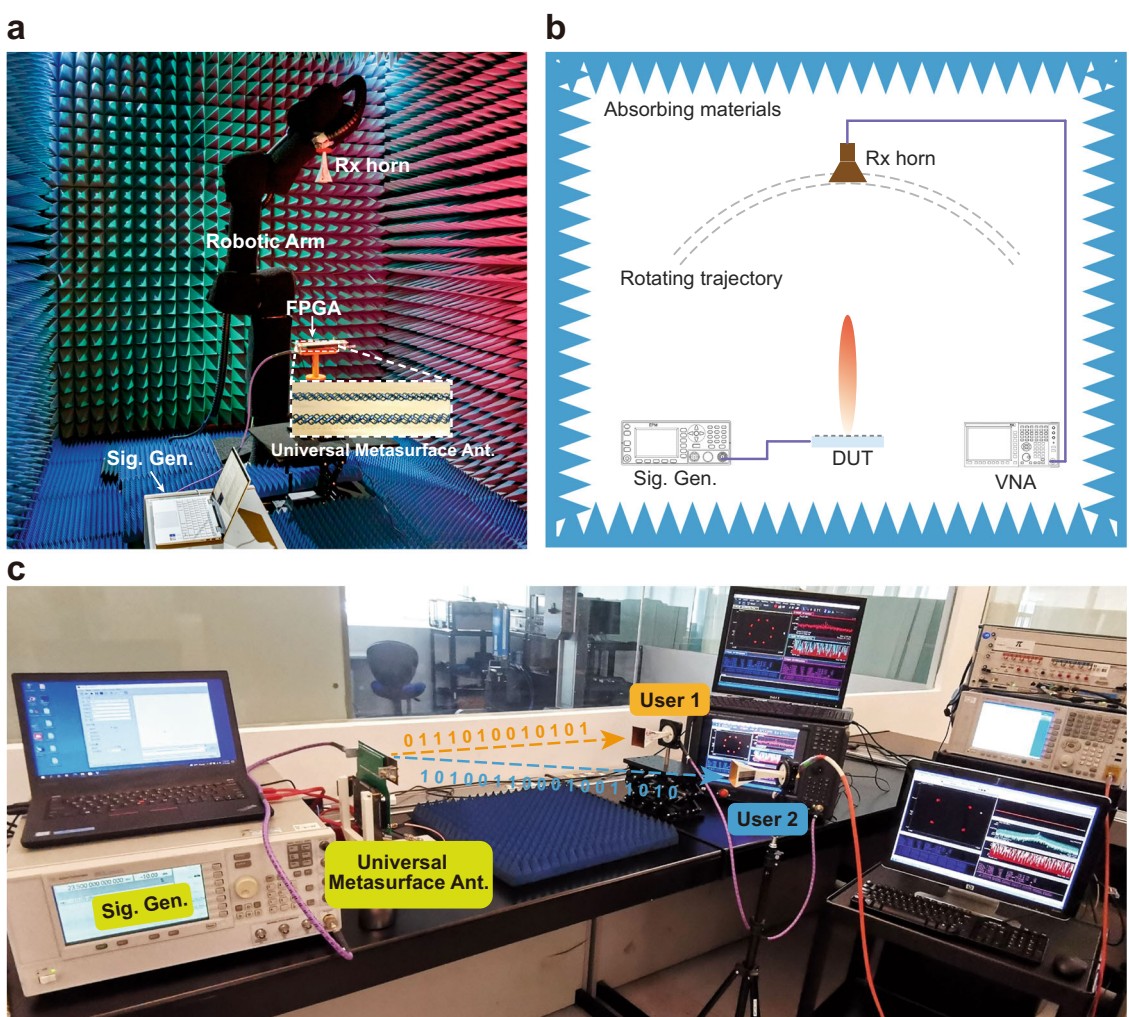

**Fig. 6 | Prototype characterization. a, b** Photograph and schematic of the measurement setup for radiation pattern measurement of the UMA. **c** Dual-channel wireless communications link testbed, in which the UMA directly and simultaneously generates two independent modulated waveforms carrying different information. Two horn antennas connected to two vector signal analyzers (VSAs) are used to receive and demodulate the signals. DUT device under test, VNA vector network analyzer, Rx receiver, Sig. Gen. signal generator.

We substitute Eq. (6) into Eq. (5) to yield:

$$\bar{\bar{m}}_i = H_0 \sum_{m=-\infty}^{\infty} e^{j2\pi(f_0+mf_M)t} \bar{\bar{p}}_{i,m} e^{-j\xi_{gw}x_i} \tag{8}$$

From Eq. (8), one can obtain that the periodic ON-OFF switching of the meta-atom will generate an infinite number of harmonic frequencies with a frequency interval $f_M$. This nonlinear effect opens the opportunity to control the frequency property of EM waves. Moreover, the spatiotemporal modulation introduces an equivalent magnetic polarizability $\bar{\bar{p}}_{i,m}(f_0 + mf_M)$ at the $m$th-order harmonic frequency, providing additional degrees of freedom to control the other properties of EM waves. Once the excited magnetic polarizability of each meta-atom is known, the radiation pattern of the UMA in free space can be obtained (Supplementary Note 1).

Firstly, we adopt the most straightforward identical rectangular time sequence (Supplementary Fig. S1c with $\Delta t = 0$) to all meta-atoms. The corresponding magnetic polarizability can be written as

$$\bar{\bar{P}}_i(t) = P_0 \begin{cases} \bar{\bar{1}} & -\bar{\bar{\tau}}_i/2 \le \bar{\bar{t}}/T_M \le \bar{\bar{\tau}}_i/2 \\ \bar{\bar{0}} & others \end{cases} \tag{9}$$

where $P_0$ is the constant magnetic polarizability as the meta-atom is in the radiating state. $\bar{\bar{\tau}}_i = \begin{bmatrix} \tau_i^u \\ \tau_i^v \end{bmatrix}$ is the duty cycle for the $|u\rangle$ and $|v\rangle$ components, which is defined as the ratio of the time a meta-atom is in the coupling state ('1') over a modulation time cycle. Substitute Eq. (9) into Eq. (7), the equivalent magnetic polarizability $\bar{\bar{p}}_{i,m}$ for the $i$th meta-atom at the $m$th-order harmonic frequency reads (See Supplementary Note 2 for the detailed derivation)

$$\bar{\bar{p}}_{i,m} = P_0 \bar{\bar{\tau}}_i sinc(\pi m \bar{\bar{\tau}}_i) \tag{10}$$

From Eq. (10), the amplitude of the magnetic polarizability at each harmonic is a function of the employed duty cycle $\bar{\bar{\tau}}_i$. To further enable phase control, we introduce a time delay $\bar{\bar{t}}_i = \begin{bmatrix} t_i^u \\ t_i^v \end{bmatrix}$ to the meta-atom (Supplementary Fig. S1c), and the equivalent magnetic polarizability $\bar{\bar{p}}_{i,m}$ becomes

$$\bar{\bar{p}}_{i,m} = P_0 \bar{\bar{\tau}}_i sinc(\pi m \bar{\bar{\tau}}_i) e^{-j2\pi m f_M \bar{\bar{t}}_i} \tag{11}$$

Comparing Eqs. (10) and (11), an additional phase term $\varphi_{ST} = -2\pi m f_M \bar{\bar{t}}_i$, depending on the applied time delay $\bar{\bar{t}}_i$, is

introduced to the equivalent magnetic polarizability at the $m$th-order harmonic frequency (Supplementary Fig. S1d). In this manner, the amplitude and phase of the excited magnetic polarizability can be decoupled and controlled independently.

## Wave manipulations without frequency shifting

For the wave manipulations at the fundamental frequency ($m = 0$), the equivalent magnetic polarizability due to the spatiotemporal modulation in Eq. (7) can be simplified as

$$\bar{p}_{i,m=0} = \frac{1}{T_M} \int_0^{T_M} \bar{\bar{P}}_i(t)dt \tag{12}$$

We can observe that the equivalent magnetic polarizability at the fundamental frequency is the time-average magnetic polarizability for one modulation cycle. Specifically, when a rectangular time sequence is adopted, the equivalent magnetic polarizability at the fundamental frequency in Eq. (11) simplifies as $\bar{p}_{i,m=0} = P_0\bar{\bar{t}}_i$. The spatiotemporal modulation generates an equivalent amplitude of the magnetic polarizability without introducing any phase shifts or momentum for the fundamental frequency[44]. To facilitate GW to PW transformation at the fundamental frequency, we leverage the equivalent magnetic polarizability to form a sinusoidal amplitude modulation along the length of the waveguide such that the $n = -1$ space harmonic becomes fast and radiates into free space (Supplementary Fig. S3a)[61]

$$\bar{\bar{A}}(x) = A_0\left[1 + \bar{\bar{M}}\cos\left(\frac{2\pi}{\Lambda}x\right)\right] \tag{13}$$

where $\bar{\bar{M}} = \begin{bmatrix} M^u \\ M^v \end{bmatrix}$ are the modulation depths for the $|u\rangle$ and $|v\rangle$ components, and $\Lambda$ is the spatial period of the sinusoidal amplitude envelope. The radiation of the equivalent magnetic dipole can be viewed as spatially sampling the reference guided wave at each meta-atom position.

The momentum of the $n = -1$ space harmonic along the $x$ direction is $k_x = \xi_{gw} - 2\pi/\Lambda$, which matches that of the free space providing that $-1 < k_x/\xi_0 < 1$. The corresponding output angle of the $n = -1$ space harmonic is $\theta_r = \sin^{-1}(\frac{\xi_{gw}-2\pi/\Lambda}{\xi_0})$[61]. Furthermore, to suppress the higher-order harmonic frequencies, we randomize the applied time delays and the time sequence yet maintain the equivalent sinusoidal modulation to the meta-atoms[44] (Supplementary Fig. S3b). Since none of the higher-order harmonic frequencies' momentums match that of the free space and waveguide, all the undesired higher-order harmonic frequencies ($m \neq 0$) are highly suppressed.

## Inherent directional modulation of the UMA

The direction-dependent phase property at the fundamental frequency can be elucidated from the perspective of spatial Fourier transform. The aperture field distribution of the metasurface and its spatial frequency spectrum (far-field radiation pattern in free space) $F(\theta)$ fulfills the Fourier transform relationship $A(x)e^{-j\xi_{gw}x} \xrightarrow{FT} F(\theta)$, where $A(x)$ is the equivalent spatial amplitude envelope in Eq. (13) imparted by the spatiotemporal modulation. Now we introduce a space translation $\Delta x$ to the spatial amplitude envelope (see Supplementary Fig. S7a). The aperture field distribution becomes $A(x - \Delta x)e^{-j\xi_{gw}x}$, whose far-field radiation pattern is (see Supplementary Note 3 for the detailed derivation)

$$F'(\theta) = e^{-j(\xi_{gw}-\xi_0\sin\theta)\Delta x}F(\theta) \tag{14}$$

We can observe from Eq. (14) that a space translation $\Delta x$ of the amplitude envelope introduces an additional phase shift $\Delta\phi(\theta) = -(\xi_{gw} - \xi_0\sin\theta)\Delta x$ to the phase pattern without affecting the power pattern of the UMA. Moreover, the introduced phase shift $\Delta\phi$ is a function of the observation direction $\theta$, meaning that different observation directions possess different phase shifts for a fixed space translation $\Delta x$ of the amplitude envelope. As a special case, the phase shift in the main beam direction $\theta_r$ is $\Delta\phi(\theta_r) = -\frac{2\pi}{\Lambda}\Delta x$, considering the main beam direction is $\theta_r = \sin^{-1}\left(\frac{\xi_{gw}-2\pi/\Lambda}{\xi_0}\right)$[61].

For clear illustration purposes, here we take the simplest BPSK modulation scheme as an example to show the directional information modulation of our UMA. Supplementary Fig. S7a shows the two equivalent sinusoidal amplitude envelopes to generate high-directivity beams with the same amplitude yet 180° phase difference at the main-beam direction ($\theta_r = 0°$), mapping to the digital information '0' and '1', respectively. To this end, the two equivalent sinusoidal amplitude envelopes share an identical spatial period $\Lambda$ yet with a space shift $\Delta x = \Lambda/2$. The phase difference between the two radiating cases as a function of the observation direction $\theta$ is presented in Supplementary Fig. S7b based on Eq. (14). We observe that a desired 180° phase difference is generated at the main beam direction ($\theta_r = 0°$ in this case), while deviating significantly from off-broadside directions. To directly generate BPSK modulated waveform, we apply the dynamic STC matrixes to the UMA according to the desired transmitted '0/1' digital stream (Supplementary Fig. S7c). Supplementary Fig. S7d further shows the theoretical radiation power pattern of the UMA (Supplementary Note 1) and the calculated error vector magnitude (EVM) at different observation directions in a noiseless environment. We observe that eavesdroppers receive a much weaker power at off-main-beam directions. Most importantly, off-angle eavesdroppers sense incorrect phase variation (information) compared to the target user in the main beam direction, leading to corrupted constellations.

We further calculate the bit error rates (BERs) for the receivers at different directions to validate the direction modulation of the UMA. The closed-form BER equation for BPSK signal in an additive white Gaussian noise (AWGN) channel can be expressed as[62]

$$P_b = Q(\sqrt{2\gamma_b}) \tag{15}$$

where $\gamma_b$ is the signal-to-noise power ratio (SNR) per bit. $Q(t)$ is the cumulative distribution function of the standard Gaussian random variable

$$Q(t) = \int_t^\infty \frac{1}{\sqrt{2\pi}}e^{-\frac{x^2}{2}}dx \tag{16}$$

The Monte Carlo simulation is widely adopted to calculate the signal BER in digital communications systems. Supplementary Fig. S7e shows the theoretical (black dot) and simulated (black line) BERs as a function of SNR. We can observe that excellent agreement between the two results, verifying the Monte Carlo simulation as an effective approach to calculating the BER. Due to the lack of a closed-form expression in BER calculation for the UMA, here we adopt the Monte Carlo simulation to calculate the BER performance in different directions. In the simulation, the power of the AWGN in different directions is identical. The simulated BERs versus SNR for receivers at 0° to 50° are shown in Supplementary Fig. S7e. Moreover, Supplementary Fig. S7f presents the simulated BER as a function of the observation direction when the SNR at the main beam direction is set as 20 dB. We can observe that receivers in the main-beam direction enjoy a much smaller BER than that of receivers at the sidelobe regions. Therefore, eavesdroppers at the sidelobe region have an extremely low probability of intercepting the information.

The inherent directional modulation of the UMA is a kind of static direction modulation[63]. An eavesdropper may still exploit the correlation between the main beam and sidelobe. However, the BER at the sidelobe region is significantly larger than that of the main beam region. An eavesdropper can easily intercept the information for conventional transmitters without any direction modulation due to the broadcast nature of wireless communications. Our UMA can significantly reduce the probability of successful data interception by eavesdroppers. The IDM property of the UMA is equally applied to other higher-order modulation formats, such as 8PSK and QAM.

## Prototype design

The configurations of the anisotropic meta-atom are illustrated in Fig. 1b. For easy integration with other components, here we adopt the SIW as the waveguiding structure, which uses two parallel rows of metallic via holes and the thin dielectric substrate to realize the rectangular waveguide in planar form. Two ±45° inclined elliptical slot openings are etched on the top metallic surface of the SIW. The lattice size of the unit cell along the $x$-direction is 2 mm, corresponding to $0.158\lambda_0$. Each slot opening works as a polarizable magnetic dipole whose extracted electric field polarization is perpendicular to the long side of the slot. The slot meta-atom is intentionally designed at an off-resonance state by carefully tuning the geometric parameters of the slot opening. Four PIN diodes (MACOMMADP-000907-14020x) are placed across the capacitive gaps of the slot openings in each meta-atom (Fig. 1b). A DC bias circuit is integrated into the meta-atom design. The circuit consists of a fan-shaped bias line (for radio-frequency choking) on the bottom biasing circuitry and a control via connecting the top meta-atom to the bottom biasing circuitry. We directly utilize the vias fences that existed in the SIW as the control vias to alleviate the perturbation of the bias network on the guided wave. Two PIN diodes in the same slot opening are biased in the same state, while PIN diodes in different inclined slot openings are biased and controlled independently. The UMA consists of 41 meta-atoms with 164 PIN diodes in this proof-of-concept example. A low-cost FPGA control board (ALTERA Cyclone IV) generates 82 independent control signals (two independent control signals for each anisotropic meta-atom) to control the ON-OFF states of the PIN diodes and the radiating states of the meta-atoms.

The proposed UMA is based on the 1-bit ON-OFF spatiotemporal modulation of the meta-atoms. Therefore, the limitation for the EM wave's property control is largely determined by the ON-OFF switch speed of the meta-atom. The adopted PIN diode (MACOMMADP-000907-14020x) can support the fastest switching speed of 2–3 ns. Although the adopted FPGA (ALTERA Cyclone IV) allows a maximum internal clock speed of 1.3 GHz using a phase-locked loop, the relatively low-speed interconnection (Supplementary Fig. S13a) between the FPGA board and metasurface antenna board limits the maximum control speed of the whole metasurface system. Supplementary Fig. S13b–d shows the measured control signal waveforms with different frequencies from the FPGA using an oscilloscope. We can observe that the signal waveform with 50 MHz distorts from a perfect rectangular waveform due to the parasitic inductance of the interconnection. Therefore, the maximum frequency shift for the current metasurface system is around 20 MHz and the transmission data rate of 2 Mbps for information transmission. Nevertheless, the limitation of the control speed of the metasurface system can be largely mitigated by adopting, e.g., (1) high-speed interconnect technologies (such as FMC connector, PCI Express), (2) packaging the FPGA and metasurface on the same PCB board, and (3) using high-speed PCB routing.

The radiation characteristics of the meta-atom are modeled and simulated using the commercially available ANSYS HFSS numerical simulator based on the finite element method. The PIN diode was modeled as a series of resistance $R = 8\,\Omega$ and inductance $L = 30\,\text{pH}$ for forward biased (State '0') and a series of capacitance $C = 0.052\,\text{pF}$ and inductance $L = 30\,\text{pH}$ for non-bias (State '1'), respectively. Wave ports are adopted to excite the fundamental $TE_{10}$ mode of the SIW waveguide.

## Prototype fabrication and characterization

A commercial multilayer printed circuit board technology is used to fabricate the UMA. The SIW and bias circuitry are made of two Rogers 5880 substrates that are 1.575 and 0.787 mm-thick, respectively. These substrates have a relative dielectric constant of 2.2 and a loss tangent of 0.0009. They are bonded by a thin Rogers 4450 F film. After this, the 164 PIN diodes are mounted on the gaps of the slot openings through reflow soldering. The radiation pattern of the UMA was measured in a microwave anechoic chamber using a reconfigurable robotic measurement system (Fig. 6a, b). A signal generator (Agilent E8267D) launches monochromatic waves at 23.5 GHz to feed the UMA. A horn antenna is connected to a vector network analyzer (VNA, Keysight N9041B) to detect the radiated EM waves from the metasurface antenna. The robotic arm holds the linearly polarized horn antenna, which can rotate along a user-defined circular path from –90° to 90° to measure the radiation pattern of the UMA. Different polarization components of the EM waves can be measured by physically rotating the linearly polarized horn antenna.

We built up an indoor wireless communication experiment to illustrate the information manipulations of our UMA (Fig. 6c). For the transmitter side, a microwave signal generator (Agilent E8267D) is adopted to generate a monochromatic wave with a frequency $f_0 = 23.5\,\text{GHz}$ to feed the UMA. The modulated waves with temporal-variant amplitude and phase properties (information) are directly generated and launched into free space by the UMA. Random binary bit streams with different modulation formats (QPSK, 8PSK, and 16QAM) are generated and mapped to FPGA's corresponding dynamic STC matrixes. For the receiver side, a linearly polarized diagonal horn antenna connected to a vector signal analyzer (VSA, Keysight N9041B) is adopted to receive and demodulate the PW radiated from the UMA. The received digital messages can be recovered by the VSA, which provides the performance of the real-time constellation diagram, eye diagram, SNR, and EVM. The distance between the UMA and the received horn antenna is -1.2 m. The modulation frequency $f_M$ is set as 0.5 MHz, and the switching speed of the PIN diodes is 10 MHz. For the multiplexing wireless communication experiment, the UMA generates two independent channels in different directions (Fig. 6c). At the receiver side, two horn antennas with $|u\rangle$ and $|u\rangle$ polarizations are located at the two main beam directions, respectively. The horns are connected to two VSAs to decode the digital message of the two channels.

The radiation efficiency, defined as the radiated power into free space to the incident power of the antenna, can be calculated by $\eta = \frac{G}{D}$[64]. The measured realized gain $G$ of the 1-D universal metasurface antenna is 9.3 dBi at 23.5 GHz based on the gain-comparison method, which uses a standard gain horn to determine the gain of the antenna under test. Due to the unavailable cylindrical near-field scanning system for 1-D antenna directivity measurement, we utilize the simulated directivity $D = 13.1\,\text{dB}$ here. The calculated radiation efficiency of the metasurface antenna is 41.7%, corresponding to the insertion loss of −3.8 dB. According to the Friis transmission equation $P_r = P_t(\frac{\lambda}{4\pi R})^2 G_t G_r$[64], the received power for the detector depends on various factors, including the input power, the free-space loss related to the distance between the transmitter and receiver, and the gains of the transmitting and receiving antennas. In the communication link setup in Fig. 4, the measured CW received power at the detector is −33.8 dBm with a distance of 1.2 m and input power of 0 dBm for the

UMA. The measured received power is −37.7 dBm in each channel for the dual-user communication link in Fig. 5.

## Data availability

The data that support the findings of this study are presented in the paper and the Supplementary Information file.

## Code availability

The codes that support the theoretical modeling of the UMA are available from the corresponding authors upon request.

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

## Acknowledgements

This work was supported by the Hong Kong Research Grants Council of the Hong Kong SAR (T42-103/16-N, C.H.C.), the Guangdong Provincial Department of Science and Technology, China (2020B1212030002, C.H.C.), the Program of Song Shan Laboratory (Included in the management of Major Science and Technology Program of Henan Province) (221100211300-02, Q.C., 221100211300-03, J.Y.D.), the National Key Research and Development Program of China (2018YFA0701904, Q.C.), the National Natural Science Foundation of China (62288101, T.J.C. and Q.C., 62201139, J.Y.D.), the 111 Project (111-2-05, T.J.C.), the Jiangsu Province Frontier Leading Technology Basic Research Project (BK20212002, T.J.C.), the Fundamental Research Funds for the Central Universities (2242022k60003, Q.C.), the National Science Foundation (NSFC) for Distinguished Young Scholars of China (62225108, Q.C.), the Southeast University - China Mobile Research Institute Joint Innovation Center (R207010101125D9, Q.C.), and the IEEE Antennas and Propagation Society (AP-S) Fellowship (G.B.W.).

## Author contributions

Q.C., T.J.C., and C.H.C. suggested the designs, planned and supervised the entire study, and led the project. G.B.W. and J.Y.D. conceived the idea of this work and designed the metasurface. G.B.W., J.Y.D., K.M.S., K.F.C. carried out the measurements and data analysis. All authors contributed to the writing of the paper. All authors discussed the theoretical modeling and numerical simulations and reviewed the manuscript.

## Competing interests

The authors declare no competing interests.
