## [Peer Review File · Nature Communications]

A universal metasurface antenna to manipulate all fundamental characteristics of electromagnetic wavesREVIEWER COMMENTS

Reviewer #1 (Remarks to the Author):

Although there have been significant progress in controlling different degrees of freedom of electromagnetic waves such as amplitude, phase, polarization, momentum, and frequency simultaneously and independently, it hasn't been demonstrated yet how to overcome this challenge for all of these degrees of freedoms. This manuscript presents experimental results to address this problem at least in the microwave regime. The concept is based on a waveguide integrated metasurface consisting of an array of meta-atom antennas. Each antenna can be independently and simultaneously addressed using FPGAs, hence leading to a plethora of wavefront engineering options ranging from beam steering, focusing, structured illumination to various division multiplexing and secure communications capabilities potentially useful for e.g., 6G communications to quantum information. The idea may also potentially extended to THz and optical frequencies in future studies. The theoretical and experimental methods are given in detail. However, the manuscript may be improved further by addressing the following comments:

1. The geometry in Fig. 1a is based on a waveguide architecture with 1D antenna array. How scalable is this type of system to a 2D antenna array with FPGAs for 2D wavefront engineering?
2. The experimentally claimed circular polarizations somewhat deviate from theoretical ones (see Supplementary Table S1). What are the reasons for this deviation? Is there any leeway for experimental improvement?
3. It would be helpful if the authors would comment on the limitations for controlling each degree of freedom in their system. Also, what are the fundamental limits besides technical ones?
4. The focusing feature of the metasurface is described by Eqs. 3 and 4 for a space coordinate $F=(x_F, z_F)$. However, these equations do not seem to include any reference to z_F . This implies focusing independent of z_F . Please clarify.
5. It's been claimed that the present metasurface can enable secure communications based on inherent directional modulation while suppressing the information leaking into sidebands. Security of a communications system usually requires formal mathematical security proofs. Is there any mathematical reasoning behind such a claim? Or is it simply a low-level comparison with conventional beam scanning? In principle, any correlations between main beam and sidebands may be still exploited by an eavesdropper depending on its significance.
6. Minor comment: some proofreading would be helpful.

Reviewer #2 (Remarks to the Author):

The authors propose and claim to have demonstrated "a universal metasurface antenna capable of dynamically, simultaneously, independently, and precisely manipulating all the constitutive properties of EM waves". Although it may appear that the authors made a very bold claim, they provide in this manuscript all the elements and the fundamental details to support this claim. Therefore, I do recommend to publish this work as is.

Reviewer #3 (Remarks to the Author):

The authors of the manuscript under consideration set out to "propose and demonstrate a universal metasurface antenna capable of dynamically, simultaneously, independently, and precisely manipulating all the constitutive properties of EM waves in a software-defined manner" as proclaimed in the abstract. The authors then proceed accordingly presenting several experiments demonstrating the independent and simultaneous control of fundamental wave parameters (frequency, propagation

direction, amplitude, phase and polarization), generation of complicated wave distributions (i.e., Airy and focused beams), and various information encoding for two-channel communications. Finally, the authors report on inherent directional modulation (IDM) phenomenon attributed to the direction-dependent phase control, demonstrating that the information can be decoded only in the direction of the main beam propagation, while being unavailable through the detection of side lobes. These demonstrations although being of unequal quality suggest indeed very interesting and appealing possibilities provided by the developed platform. The paper is written well, with all essential results presented clearly, physical arguments and experiments described in sufficient detail.

There are however a few critical comments that the authors should consider and act upon.

1. The presentation is somewhat misleading with respect to the EM wavelength range that is considered in the paper. The authors spell out that their "universal metasurface antenna" operates at microwave frequencies only on p. 5, in the beginning of Section "Full manipulation of all fundamental EM-wave characteristics", i.e., AFTER the introduction presenting the motivation for the reported research and formulation of the main demonstrated results. Considering that many of the references from the introduction are to the metasurfaces operating in the optical domain, this delay in informing on the actual operating wavelength range is crucial, resulting in confusion and frustration. The reason is that there is an enormous difference in possibilities for practical realization of general EM metasurfaces in optical and microwave frequency domains. For example, subwavelength-sized PIN diodes constituting the backbone of the considered "universal metasurface antenna" do NOT exist in the optical domain. Even more misleading is the use of "light" when describing "Space-varying wave properties for complicated wave manipulations", for example, "light focusing" in the caption to Fig. 3. The whole manuscript should, in my opinion, be critically checked and corrected, starting from the very beginning and making it clear that the work is concerned with microwaves. In this respect, the Extended Data Table should be complemented with a column specifying the operating wavelength range – obviously, something that is readily available in microwaves, such as subwavelength-sized PIN diodes, is not in the optical domain, leading to insurmountable problems.

2. The authors should critically assess the requirements for realizing the demonstrated functionalities at the sufficiently high level required from the corresponding practical devices. It is, for example, clear that the beam shaping performance demonstrated in Fig. 3 can be accepted only at the level of proof-of-principle demonstration - the beams shown in Figs. 3c and 3g are fading away already at distances of 100 mm, which is less than 10 wavelengths (the free-space wavelength at 23.5 GHz is around 13 mm). One conjectures that such a drastic deterioration is related to a very large beam divergence out of the considered x-z plane, probably because of using of only one metasurface cell along the y-direction. All these very important for practical use matters should explicitly be brought up by the authors, leaving as little room for guessing as possible. Importantly, the authors should provide the information on the insertion loss – it is very important, for example, to know how much of the input CW power reaches the detector in the experiments in various information encoding schemes reported in Fig. 4. The same information should also be provided for the experiments with two information channels reported in Fig. 5.

While the demonstrations presented in the paper can be accepted as the proof-of-principle, the authors should be much more open regarding the possible practical implementation of the proposed configuration. The fabricated configuration does not merit the level of prototype, simply because the current design can hardly be used in practice. Meanwhile, the extension to a practical two-dimensional case would require the use of 26896 PIN diodes, whose individual addressing might create additional problems, especially with respect to the operation bandwidth.

The aforementioned issues have to be explicitly and in detail addressed in the revised manuscript to back up the claims of "significant merits, including full-dimensional wave controllability, inherent IDM, simplified coding scheme (1-bit), free of sideband pollution, and potential on-chip integration, making it an appealing enabler for the next-generation large-capacity and high-security information systems".

3. Given the matters and problems mentioned above the last sentence stating that "the proposed concept can be extended to the terahertz and optical ranges, leading to widespread applications such as augmented reality, holography, integrated sensing and communications for 6G, quantum optics, and quantum information science" should be either deleted or backed up with considerations of how this extending "to the terahertz and optical ranges" can be implemented in practice, since the latter is

very far from being straightforward. Note that the demonstrated transmission data rates of 1 and 2 Mbps are far too low for the information communication systems that are already in use, making it difficult to share enthusiasm of the authors with respect to the potential of the proposed metasurface configuration for future use.

4. Minor issues: (a) the references used in the first sentence "Electromagnetic (EM) waves, ranging from microwave and terahertz waves to visible light, are the bases of various disciplines ranging from optics ..." seem to be out place, being not general (e.g., not about optics as such) but very specifically dealing with metasurfaces and metamaterials; (b) "universal metasurface antenna" is found in 71 places of the main text, meriting definitely to abbreviate, while other abbreviations should be checked whether their use is reasonable or not (using 2-3 times is not sufficient, in my opinion).

Responses to Reviewers' Comments

The authors would like to thank the editors and reviewers for their time and constructive comments to improve the manuscript. The replies to all comments are provided below in blue colors. The revised portions in the manuscript are highlighted in yellow background.

Responses to Reviewer #1:

Comment:

Although there have been significant progress in controlling different degrees of freedom of electromagnetic waves such as amplitude, phase, polarization, momentum, and frequency simultaneously and independently, it hasn't been demonstrated yet how to overcome this challenge for all of these degrees of freedoms. This manuscript presents experimental results to address this problem at least in the microwave regime. The concept is based on a waveguide integrated metasurface consisting of an array of meta-atom antennas. Each antenna can be independently and simultaneously addressed using FPGAs, hence leading to a plethora of wavefront engineering options ranging from beam steering, focusing, structured illumination to various division multiplexing and secure communications capabilities potentially useful for e.g., 6G communications to quantum information. The idea may also potentially extended to THz and optical frequencies in future studies. The theoretical and experimental methods are given in detail. However, the manuscript may be improved further by addressing the following comments:

Response:

Thank you very much for your appreciation, comprehensive review, and constructive comments. We have carefully revised and improved the manuscript and responded to your comments provided below.

Comment:

1. The geometry in Fig. 1a is based on a waveguide architecture with 1D antenna array. How scalable is this type of system to a 2D antenna array with FPGAs for 2D wavefront engineering?

Response:

We thank the referee for pointing out this important issue for the scalability to a 2-D antenna array. One solution for implementing a 2-D metasurface antenna is by periodically repeating the 1-D metasurface antenna along the y-axis, fed by a power-dividing network, as shown in Fig. R1a. In this case, eight identical 1-D SIWs arrays with a total of 328 meta-atoms and 1312 PIN diodes form a 2-D radiating aperture. A similar topology for SIW-based 2-D antennas has been demonstrated in refs. [R1]-[R2], although without any spatiotemporal modulation. Fig. R1b shows the configuration of the 8-way SIW parallel feeding divider and its simulated E-field distribution at 23.5 GHz using a commercially available ANSYS HFSS numerical simulator. It can be observed that the treelike power divider can divide the input power into eight parallel ways with the same amplitude and phase to feed the metasurface antenna array.

Fig. R1 | **a**, Configuration of the 2-D universal metasurface antenna. **b**, Simulated E-field distribution for the 8-way power divider at 23.5 GHz.

Fig. R2 | 2-D universal metasurface antenna for 2-D wavefront engineering. **a**, 2-D STC matrix for each 1-D metasurface array to steer the beam to the direction $(\theta, \varphi) = (30^\circ, 45^\circ)$. **b-c**, The corresponding 3-D STC matrix (b) and its calculated radiation patterns in uv-space at different harmonic frequencies. (c). **d-f**, Calculated 3-D radiation patterns at the 1st harmonic frequency as the main beam scans to $(\theta, \varphi) = (30^\circ, 45^\circ)$, $(0^\circ, 0^\circ)$ and $(30^\circ, -135^\circ)$, respectively.

The 2-D metasurface antenna can realize 2-D wavefront engineering by extending the original 2-D STC matrix into a 3-D one. We show 2-D beam scanning for the 1st harmonic radiation without loss of generality. Firstly, each 1-D metasurface antenna has its 2-D STC matrix to control the wavefront along the x-direction. We have demonstrated in Fig. 1e of the main text that the reference time shift can control the phase of the radiated wave for the 1-D metasurface antenna. To enable the beam also steer in the y-direction, we apply different reference time shifts to the eight 1-D metasurface antennas to form a progressive phase shift along the y-direction. The corresponding eight different 2-D STC matrixes to the eight 1-D metasurface antennas for beam steering to $(\theta, \varphi) = (30^\circ, 45^\circ)$ are shown in Fig. R2a. Cascading these eight 2D STC matrixes forms a 3D STC matrix, as shown in Fig. R2b. Fig. R2c shows the

corresponding theoretical radiation patterns in uv-space at different harmonics. The 3-D radiation pattern at the 1st harmonic frequency is presented in Fig. R2d. The radiation patterns are calculated based on the theoretical model for STC metasurface antennas [44]. We extend the 1-D spatial Fourier transform in Eq. (10) in ref. [44] to the 2-D spatial Fourier transform to calculate the 3-D radiation pattern here. We can observe that a high-directivity beam radiates to the desired direction at the 1st harmonic frequency. Again, other undesired harmonics are highly suppressed due to the phase mismatch in each SIW waveguide and free space.

As proof-of-concept examples, the 2-D universal metasurface antenna is designed to steer the beam to $(\theta, \varphi) = (30^\circ, 45^\circ)$, $(0^\circ, 0^\circ)$ and $(30^\circ, -135^\circ)$, respectively. The theoretically calculated 3-D radiation patterns at the 1st harmonic frequency are shown in Figs. R2d-f. We can observe that the output beam of the 2-D metasurface antenna can be correctly pointed in the intended direction. These results verify the 2-D wavefront engineering capability of the 2-D metasurface antenna.

We have finished the design of the 2-D metasurface antenna. Experimental results will be given in a future paper when the measurement results are available. The contents for extending to a 2-D metasurface antenna have been added as Supplementary Note 4 of the revised manuscript.

[R1] Cheng, Y. J., Hong, W. & Wu, K. 94 GHz substrate integrated monopulse antenna array. *IEEE Trans. Antennas Propag.* 60, 121-129 (2011).

[R2] Wu, Y. F., Cheng, Y. J. & Huang, Z. X. Ka-band near-field-focused 2-D steering antenna array with a focused Rotman lens. *IEEE Trans. Antennas Propag.* 66, 5204-5213 (2018).

Comment:

2. The experimentally claimed circular polarizations somewhat deviate from theoretical ones (see Supplementary Table S1). What are the reasons for this deviation? Is there any leeway for experimental improvement?

Response:

Thank you very much for the valuable comments. The discrepancy between the theoretical and experimental results is mainly caused by the weak coupling between the two $\pm 45^\circ$ -inclined slot openings in each meta-atom. To investigate the coupling effects,

Figs. R3a,b show two meta-atom configurations with and without the coupling from the other slot opening. Meta-atom I with only one $+45^\circ$ -inclined slot opening on the top of the SIW waveguide; Meta-atom II with two $\pm 45^\circ$ -inclined slot openings, which is the configuration used in the main text and in the experimental demonstration. In both cases, the PIN diodes in the $+45^\circ$ -inclined slot-opening are in the OFF state such that the meta-atom radiates $|v\rangle$ (-45° from the x-axis) linear polarization. In Meta-atom II, the PIN diodes in the -45° -inclined slot opening are in the ON state. Fig. R3d shows the simulated co- and cross-polarized radiation patterns for the two cases using the full-wave simulator ANSYS HFSS. It can be observed that the co-polarized patterns are almost identical, while the cross-polarization level in the broadside direction ($\theta = 0^\circ$) increases from -20.9 dB to -17.9 dB for Meta-atom II with the existence of the -45° -inclined slot opening. This is due to the fact that the -45° -inclined slot opening still contributes to the cross-polarization radiation, although the PIN diodes are in the ON state. In the theoretical model developed in the main text, the weak spatiotemporal couplings between the two slot openings are not considered, causing the deviation from the measurement results.

Fig. R3 | Coupling effects of the two $\pm 45^\circ$ -inclined slot openings in each meta-atom. **a**, Meta-atom configuration with only one $+45^\circ$ -inclined slot opening on the top of the SIW waveguide. **b**, Meta-atom configuration with two $\pm 45^\circ$ -inclined slot openings. **c**, Meta-atom configuration with two $\pm 45^\circ$ -inclined slot openings; each slot opening has four PIN diodes to control its radiation state. **d**, Simulated co-pol (-45° linear polarized)

and cross-pol (+45° linear polarized) radiation patterns for different meta-atom configurations.

The coupling between the two slot openings can be significantly suppressed by shifting the resonant frequency of the -45°-inclined slot opening far away from the operating frequency. One solution is to adopt four PIN diodes at each slot opening, as shown in Meta-atom III in Fig. R3c. The equivalent magnetic current path of the -45°-inclined slot opening is reduced, shifting the resonant frequency to a higher frequency compared to that of Meta-atom II. Fig. R3d shows the simulated radiation patterns of the Meta-atom III. We can observe that the cross-polarization is similar to that of Meta-atom I, indicating the coupling from the -45°-inclined slot opening is negligible for Meta-atom III.

The cause of the discrepancy between the theoretical and experimental results and its improvement solution have been added as Supplementary Note 5 of the revised manuscript.

Comment:

3. It would be helpful if the authors would comment on the limitations for controlling each degree of freedom in their system. Also, what are the fundamental limits besides technical ones?

Response:

Thank you very much for the insightful comments. The proposed universal metasurface antenna is based on the 1-bit ON-OFF spatiotemporal modulation of the meta-atoms. Therefore, the limitation for the EM wave's property control is largely determined by the ON-OFF switch speed of the meta-atom. The adopted PIN diode (MACOMMADP-000907-14020x) can support the fastest switching speed of 2-3 ns. Although the adopted FPGA (ALTERA Cyclone IV) allows a maximum internal clock speed of 1.3 GHz using a phase-locked loop, the relatively low-speed interconnector (Fig. R4a) between the FPGA board and metasurface antenna board limits the maximum control speed of the whole metasurface system. Figs. R4b-d show the measured control signal waveforms with different frequencies from the FPGA using an oscilloscope. We can observe that the signal waveform with 50 MHz distorts from a perfect rectangular waveform due to the parasitic inductance of the interconnector. Therefore, the

maximum frequency shift for the current metasurface system is around 20 MHz, and the transmission data rate of 2 Mbps for information transmission.

Fig. R4. | **a**, Photograph of the interconnect between the FPGA board and metasurface antenna board. **b-c**, Measured control signal waveforms from FPGA with frequencies of 10 MHz (b), 20 MHz (c), and 50 MHz (d) using an oscilloscope.

We want to emphasize that the current universal metasurface antenna is a proof-of-concept example demonstrating its flexibility for complete control of the five fundamental wave properties and direct information transmission, although not at a high-speed data rate. Furthermore, the limitation of the control speed of the metasurface system can be largely mitigated by adopting, e.g., 1) high-speed interconnect technologies (such as FMC connector, PCI Express), 2) packaging the FPGA and metasurface on the same PCB board, and 3) using high-speed PCB routing.

The limitation of the current metasurface antenna has been added on Pages 21-22 and Supplementary Fig. S13 of the revised manuscript.

Comment:

4. The focusing feature of the metasurface is described by Eqs. 3 and 4 for a space coordinate $F=(x_F, z_F)$. However, these equations do not seem to include any reference to z_F . This implies focusing independent of z_F . Please clarify.

Response:

Sorry for the confusion, and thank you very much for pointing out this important issue. The “ z_F ” in Eqs. (3) and (4) were mistakenly written as “ x_F ” in the original main text.

We apologize for any confusion or inconvenience that this may have caused. The corrected Eqs. (3) and (4) are:

$$k_x(x) = \xi_m \frac{x_F - x}{\sqrt{(x_F - x)^2 + (z_F - z)^2}} \quad (\text{R1})$$

$$\partial t_i(x)/\partial x = [\xi_m \frac{x_F - x}{\sqrt{(x_F - x)^2 + (z_F - z)^2}} - \xi_{gw}] / (2\pi m f_M) \quad (\text{R2})$$

The required linear momentum for wave focusing is inferred based on the geometric ray approach and the geometrical relationship, as shown in Fig. R5. For each meta-atom position $(x, z = 0)$, the output angle should satisfy

$$\sin\theta = \frac{x_F - x}{\sqrt{(x_F - x)^2 + (z_F - z)^2}} \quad (\text{R3})$$

The linear momentum along the x -direction should be

$$\sin\theta = \frac{k_x}{\xi_m} \quad (\text{R4})$$

Combining Eqs. (R3) and (R4), Eq. (R1) can be obtained.

Fig. R5 | Geometric representation of the wave-focusing for the universal metasurface antenna.

Note that the results in Fig. 3e and Figs. 3f-g in the original manuscript are obtained based on the corrected Eqs. (R1) and (R2). We have updated the corrected Eqs. (3) and (4) in the revised manuscript.

Comment:

5. It's been claimed that the present metasurface can enable secure communications based on inherent directional modulation while suppressing the information leaking into sidebands. Security of a communications system usually requires formal mathematical security proofs. Is there any mathematical reasoning behind such a claim? Or is it simply a low-level comparison with conventional beam scanning? In principle, any correlations between main beam and sidebands may be still exploited by an

eavesdropper depending on its significance.

Response:

Thank you very much for your constructive comments. Firstly, the inherent directional modulation is due to the direction-dependent phase control of the universal metasurface antenna. The phase information can only be correctly coded in the main beam direction, while being erroneously coded in sidelobe regions. The detailed mathematical derivation of the direction-dependent phase property of the universal metasurface antenna has been given in Methods and Supplementary Note 3 of the original manuscript. Secondly, as per your suggestion, we further calculate the bit error rates (BERs) for the receivers at different directions to prove the direction modulation of the universal metasurface antenna. We take the simplest BPSK modulation scheme as an illustrative example. The closed-form BER equation for the BPSK signal in an additive white Gaussian noise (AWGN) channel can be expressed as [R3]

$$P_b = Q(\sqrt{2\gamma_b}) \quad (\text{R5})$$

where γ_b is the signal-to-noise power ratio (SNR) per bit. $Q(t)$ is the cumulative distribution function of the standard Gaussian random variable, i.e.,

$$Q(t) = \int_t^{\infty} \frac{1}{\sqrt{2\pi}} e^{-\frac{x^2}{2}} dx \quad (\text{R6})$$

The Monte Carlo simulation is widely adopted to calculate the signal BER in digital communications systems. Fig. R6a shows the theoretical (black dot) and simulated (black line) BERs as a function of SNR. We can observe that excellent agreement between the two results, verifying the Monte Carlo simulation as an effective approach to calculating the BER. Due to the lack of a closed-form expression in BER calculation for the universal metasurface antenna, here we adopt the Monte Carlo simulation to calculate the BER performance in different directions. In the simulation, the power of the AWGN in different directions is identical. The simulated BERs versus SNR for receivers at 0° to 50° are shown in Fig. R6a. Moreover, Fig. R6b presents the simulated BER as a function of observation direction when the SNR at the main beam direction is set as 20 dB. We can observe that receivers in the main-beam direction enjoy a much smaller BER than that of receivers at the sidelobe regions. Therefore, eavesdroppers in the sidelobe regions have an extremely low probability of intercepting the information.

Fig. R6 | **a**, BER versus SNR for a receiver in different directions of the universal metasurface antenna. **b**, Calculated BER as a function of the observation direction based on the Monte Carlo simulation. The SNR at the main beam direction ($\theta = 0^\circ$) is set as 20 dB.

The inherent directional modulation of the universal metasurface antenna is a kind of static direction modulation [R4]. We agree that an eavesdropper may still exploit the correlation between the main beam and sideband. However, the BER at the sidelobe region is significantly larger than that of the main beam region. An eavesdropper can easily intercept the information for conventional transmitters without any direction modulation due to the broadcast nature of wireless communications. Our universal metasurface antenna can significantly reduce the probability of successful data interception by eavesdroppers.

The BER performance of the universal metasurface antenna has been added on Pages 19-20 of the revised manuscript.

[R3] Goldsmith, A. Wireless communications. (Cambridge university press, 2005).

[R4] Ding, Y. & Fusco, V. F. Establishing metrics for assessing the performance of directional modulation systems. *IEEE Trans. Antennas Propag.* 62, 2745-2755 (2014).

Comment:

6. Minor comment: some proofreading would be helpful.

Response:

Thank you very much for the suggestion. We have carefully proofread the manuscript with the help of a native English speaker.

Responses to Reviewer #2

Comment:

The authors propose and claim to have demonstrated "a universal metasurface antenna capable of dynamically, simultaneously, independently, and precisely manipulating all the constitutive properties of EM waves". Although it may appear that the authors made a very bold claim, they provide in this manuscript all the elements and the fundamental details to support this claim. Therefore, I do recommend to publish this work as is.

Response:

We thank the referee for the positive feedback and the kind support.

Responses to Reviewer #3

Comment:

The authors of the manuscript under consideration set out to “propose and demonstrate a universal metasurface antenna capable of dynamically, simultaneously, independently, and precisely manipulating all the constitutive properties of EM waves in a software-defined manner” as proclaimed in the abstract. The authors then proceed accordingly presenting several experiments demonstrating the independent and simultaneous control of fundamental wave parameters (frequency, propagation direction, amplitude, phase and polarization), generation of complicated wave distributions (i.e., Airy and focused beams), and various information encoding for two-channel communications. Finally, the authors report on inherent directional modulation (IDM) phenomenon attributed to the direction-dependent phase control, demonstrating that the information can be decoded only in the direction of the main beam propagation, while being unavailable through the detection of side lobes. These demonstrations although being of unequal quality suggest indeed very interesting and appealing possibilities provided by the developed platform. The paper is written well, with all essential results presented clearly, physical arguments and experiments described in sufficient detail.

There are however a few critical comments that the authors should consider and act upon.

Response:

Thank you very much for your appreciation, comprehensive review, and kind support. We have carefully revised and improved the manuscript and responded to your comments below.

Comment:

1. The presentation is somewhat misleading with respect to the EM wavelength range that is considered in the paper. The authors spell out that their “universal metasurface antenna” operates at microwave frequencies only on p. 5, in the beginning of Section “Full manipulation of all fundamental EM-wave characteristics”, i.e., AFTER the introduction presenting the motivation for the reported research and formulation of the main demonstrated results. Considering that many of the references from the introduction are to the metasurfaces operating in the optical domain, this delay in informing on the actual operating wavelength range is crucial, resulting in confusion and frustration. The reason is that there is an enormous difference in possibilities for

practical realization of general EM metasurfaces in optical and microwave frequency domains. For example, subwavelength-sized PIN diodes constituting the backbone of the considered “universal metasurface antenna” do NOT exist in the optical domain. Even more misleading is the use of “light” when describing “Space-varying wave properties for complicated wave manipulations”, for example, “light focusing” in the caption to Fig. 3.

The whole manuscript should, in my opinion, be critically checked and corrected, starting from the very beginning and making it clear that the work is concerned with microwaves. In this respect, the Extended Data Table should be complemented with a column specifying the operating wavelength range – obviously, something that is readily available in microwaves, such as subwavelength-sized PIN diodes, is not in the optical domain, leading to insurmountable problems.

Response:

Thank you very much for these constructive comments and suggestions, which are very helpful in improving our paper. According to your suggestions, we have modified the manuscript as follows:

- (1) We have carefully checked and corrected the whole manuscript to declare the introduced universal metasurface antenna is operating at microwave frequency, from the Abstract, Introduction, Results, Methods, to Conclusion sections.
- (2) The word “light focusing” has been modified as “wave focusing” in the “Space-varying wave properties” section and Fig. 3 of the revised manuscript.
- (3) We have added a column in Extended Data Table I to specify the operating frequency/wavelength for each metasurface for fair comparison purposes.

Comment:

2. The authors should critically assess the requirements for realizing the demonstrated functionalities at the sufficiently high level required from the corresponding practical devices. It is, for example, clear that the beam shaping performance demonstrated in Fig. 3 can be accepted only at the level of proof-of-principle demonstration - the beams shown in Figs. 3c and 3g are fading away already at distances of 100 mm, which is less than 10 wavelengths (the free-space wavelength at 23.5 GHz is around 13 mm). One conjectures that such a drastic deterioration is related to a very large beam divergence out of the considered x-z plane, probably because of using of only one metasurface cell along the y-direction. All these very important for practical use matters should

explicitly brought up by the authors, leaving as little room for guessing as possible.

Response:

Thank you very much for your valuable comments. The Airy beam and focused beam in Fig. 3 are in the Fresnel near-field zone of the antenna. The Airy and focused beams fade away at a certain distance due to the diffraction of EM waves, a common phenomenon for both 1-D and 2-D radiating apertures. For example, in our recent work in ref. [R5], we utilized a **2D** meta-lens aperture with a diameter of 30 mm ($30\lambda_0$ at 300 GHz) for Airy beam generation. The measured Airy beam fades away at around 90 mm, as shown in Fig. R7a. The focused beam generated by another metalens with the same aperture size also fades away at around 70 mm, 60 mm, and 50 mm for different focusing positions (Fig. R7b). The fading phenomena for Airy and focused beams are found in other 1-D and 2-D radiating apertures in the open literature [R6]-[R8].

Fig. R7 | Observation of Airy beam and focused beam fading phenomenon for 2-D metasurface aperture. **a**, Metasurface doublet for Airy beam generation and its measured E-field distribution on the longitudinal plane. **b**, Metasurface triplet for focused beam generation and its E-field distributions on the longitudinal plane for different focusing positions. (Reproduced from ref. [R5]).

[R5] Zhang, J. C., Wu, G. B., Chen, M. K., Liu, X., Chan, K. F., Tsai, D. P., & Chan, C. H. A 6G meta-device for 3D varifocal. *Sci. Adv.* **9**, eadf8478 (2023).

[R6] Xi, K. *et al.* Terahertz Airy beam generated by Pancharatnam-Berry phases in guided wave-driven metasurfaces. *Opt. Express* **30**, 16699-16711 (2022).

[R7] Wen, J. et al. All-dielectric synthetic-phase metasurfaces generating practical airy beams. *ACS Nano* **15**, 1030-1038 (2021).

[R8] Song, E. Y. et al. Compact generation of airy beams with C-aperture metasurface. *Adv. Opt. Mater.* **5**, 1601028 (2017).

Fig. R8 | 1-D and 2-D aperture effects on the fading distance. **a-c**, The aperture phase distribution for the 1-D metasurface antenna and its corresponding simulated E-field intensity distributions in the longitudinal plane (b) and transversal plane (c). **d-f**, The aperture phase distribution for the 2-D metasurface antenna and its corresponding simulated E-field intensity distributions in the longitudinal plane (e) and transversal plane (f).

To investigate the 1-D and 2-D antenna aperture effect on the fading distance along the z-direction, Figs. R8a,d show the aperture phase distributions for the 1-D and 2-D antennas with an intended focus spot $(x_F, y_F, z_F) = (0, 0, 80)$ mm. The radiated E-field distributions on the longitudinal (xz-plane) and transversal (xy-plane) planes calculated by the Fresnel diffraction approach [R9] assuming a uniform aperture amplitude distribution are shown in Fig. R8. From Figs. R8c,f, we can observe that the 1-D antenna has a much broader beamwidth along the y-direction than that of the 2-D antenna since there is only one meta-atom along the y-direction. Nevertheless, the 1-D and 2-D aperture antennas share a similar field distribution in the longitudinal plane (xz-plane). As shown in Figs. R8b,e, both the beams fade away at around 100 mm. Therefore, the 1-D radiating aperture only affects the field distribution along the y-direction, with negligible effects on the fading distance along the z-direction.

For the focused beam, the desired focused point determines the fading distance. Fig. R9 shows the calculated E-field distributions for three different focus spot positions $(x_F, y_F,$

$z_F = (0, 0, 60 \text{ mm}), (0, 0, 80 \text{ mm}),$ and $(0, 0, 100 \text{ mm})$. The fading distance of the 1-D metasurface antenna increases with the increase of the focus spot distance. Similarly, the fading distance of the Airy beam is determined by the antenna aperture size and the acceleration factor α in Eq. (1).

Fig. R9 | Simulated E-field intensity distributions in the longitudinal plane (xz-plane) when the accessible distance is set to 60 (a), 80 (b), and 100 mm (c), respectively.

The explanation of the fading phenomenon for the Airy and focused beams has been added in Supplementary Note 6 of the revised manuscript.

[R9] Goodman, J.W. Introduction to Fourier Optics (Roberts & Co. Publishers, Englewood, Colorado, 2005).

Comment:

Importantly, the authors should provide the information on the insertion loss – it is very important, for example, to know how much of the input CW power reaches the detector in the experiments in various information encoding schemes reported in Fig. 4. The same information should also be provided for the experiments with two information channels reported in Fig. 5.

Response:

Thank you very much for your valuable comments. The radiation efficiency, defined as the radiated power into free space to the incident power of the antenna, can be calculated by $\eta = \frac{G}{D}$ [R10]. The measured realized gain G of the 1-D universal metasurface antenna is 9.3 dBi at 23.5 GHz based on the gain-comparison method, which uses a standard gain horn to determine the gain of the antenna under test. Due to the unavailable cylindrical near-field scanning system for 1-D antenna directivity measurement in our lab, we utilize the simulated directivity $D = 13.1 \text{ dB}$ here. The calculated radiation efficiency of the metasurface antenna is 41.7%, corresponding to the insertion loss of -3.8 dB.

According to the Friis transmission equation $P_r = P_t \left(\frac{\lambda}{4\pi R}\right)^2 G_t G_r$ [R10], the received power for the detector depends on various factors, including the input power, the free-space loss related to the distance between the transmitter and receiver, and the gains of the transmitting and receiving antennas. In our communication link setups in Figs. 4, the measured CW received power at the detector is -33.8 dBm with a distance of 1.2 mm and input power of 0 dBm. The measured received power for each channel is -37.7 dBm for the dual-user communication link in Fig. 5.

This information has been added to Pages 23-24 of the received manuscript.

[R10] Balanis, C. A. Antenna theory: analysis and design. (John Wiley & sons, 2015).

Comment:

While the demonstrations presented in the paper can be accepted as the proof-of-principle, the authors should be much more open regarding the possible practical implementation of the proposed configuration. The fabricated configuration does not merit the level of prototype, simply because the current design can hardly be used in practice. Meanwhile, the extension to a practical two-dimensional case would require the use of 26896 PIN diodes, whose individual addressing might create additional problems, especially with respect to the operation bandwidth.

The aforementioned issues have to be explicitly and in detail addressed in the revised manuscript to back up the claims of “significant merits, including full-dimensional wave controllability, inherent IDM, simplified coding scheme (1-bit), free of sideband pollution, and potential on-chip integration, making it an appealing enabler for the next-generation large-capacity and high-security information systems”.

Response:

Thank you very much for raising this important question. The universal metasurface antenna was practically implemented by standard commercial printed circuit board (PCB) technology. More information on the universal metasurface antenna fabrication is provided on Page 22 of the revised manuscript.

One solution for extending to a 2-D metasurface antenna is by periodically repeating the 1-D metasurface antenna along the y-axis, fed by a power-dividing network, as shown in Fig. R10a. In this design, eight identical 1-D SIWs arrays with a total of 328

meta-atoms and 1312 PIN diodes form a 2-D radiating aperture. A similar topology for SIW-based 2-D antennas has been demonstrated in refs. [R11]-[R12], although without spatiotemporal modulation. It is important to point out that the array distance along the y-axis is not necessary to be subwavelength since the guided waves in each SIW waveguide only propagate along the x-axis. In this design, the array distance along the y-axis is 6.5 mm (around $0.5\lambda_0$ at 23.5 GHz). Thereby, the required PIN diode number for the 2D aperture can be significantly reduced. Moreover, individual addressing over 10000 PIN diodes/Meta-atoms for microwave metasurfaces has been reported [R13]-[R14].

[R11] Cheng, Y. J., Hong, W. & Wu, K. 94 GHz substrate integrated monopulse antenna array. *IEEE Trans. Antennas Propag.* 60, 121-129 (2011).

[R12] Wu, Y. F., Cheng, Y. J. & Huang, Z. X. Ka-band near-field-focused 2-D steering antenna array with a focused Rotman lens. *IEEE Trans. Antennas Propag.* 66, 5204-5213 (2018).

[R13] Pan, X., Yang, F., Xu, S. & Li, M. A 10240-element reconfigurable reflectarray with fast steerable monopulse patterns. *IEEE Trans. Antennas Propag.* 69, 173-181 (2020).

[R14] Kamoda, H., Iwasaki, T., Tsumochi, J., Kuki, T. & Hashimoto, O. 60-GHz electronically reconfigurable large reflectarray using single-bit phase shifters. *IEEE Trans. Antennas Propag.* 59, 2524-2531 (2011).

Fig. R10 | **a**, Configuration of the 2-D universal metasurface antenna. **b**, Simulated E-field distribution for the 8-way power divider at 23.5 GHz.

Fig. R10b shows the configuration of the 8-way SIW parallel feeding divider and its simulated E-field distribution at 23.5 GHz using a commercially available ANSYS HFSS numerical simulator. We can observe that the treelike power divider can divide the input power into eight parallel ways with the same amplitude and phase to feed the metasurface antenna array.

The 2-D metasurface antenna can realize 2-D wavefront engineering by extending the original 2-D STC matrix into a 3-D one. We demonstrate the 2-D beam scanning for the 1st harmonic radiation without loss of generality. Firstly, each 1-D metasurface antenna has its 2-D STC matrix to control the wavefront along the x-direction. We have demonstrated in Fig. 1e of the main text that the reference time shift can control the phase of the radiated wave for the 1-D metasurface antenna. To enable beam steering also in the y-direction, we apply different reference time shifts to the eight 1-D metasurface antennas to form a progressive phase shift along the y-direction. The corresponding eight different 2-D STC matrixes for the eight 1-D metasurface antennas for beam steering to $(\theta, \varphi) = (30^\circ, 45^\circ)$ are shown in Fig. R11a. Cascading these eight 2-D STC matrixes forms a 3D STC matrix, as shown in Fig. R11b. Fig. R11c shows the corresponding theoretical radiation patterns in uv-space at different harmonic frequencies. The 3-D radiation pattern at the 1st harmonic frequency is presented in Fig. R11d. The radiation patterns are calculated based on the theoretical model for STC metasurface antennas [44]. We extend the 1-D spatial Fourier transform in Eq. (10) in ref. [44] to the 2-D spatial Fourier transform to calculate the 3-D radiation pattern here. We can observe from Fig. R11c that a high-directivity beam radiates to the desired direction at the 1st harmonic frequency. Again, other undesired harmonics are highly suppressed due to the phase mismatch in each SIW waveguide and free space.

Fig. R11 | 2-D universal metasurface antenna for 2-D wavefront engineering. **a**, 2-D STC matrixes of each 1-D metasurface array for beam radiating in the direction $(\theta, \varphi) = (30^\circ, 45^\circ)$. **b-c**, The corresponding 3-D STC matrix (b) and its calculated radiation patterns in uv-space at different harmonic frequencies. (c). **d-f**, Calculated 3-D radiation patterns at the 1st harmonic frequency as the main beam scans to $(\theta, \varphi) = (30^\circ, 45^\circ)$, $(0^\circ, 0^\circ)$ and $(30^\circ, -135^\circ)$, respectively.

As proof-of-concept examples, the 2-D universal metasurface antenna is designed to steer the beam to $(\theta, \varphi) = (30^\circ, 45^\circ)$, $(0^\circ, 0^\circ)$ and $(30^\circ, -135^\circ)$, respectively. The theoretically calculated 3-D radiation patterns at the 1st harmonic frequency are shown in Fig. R11d-f. We can observe that the output beams of the 2-D metasurface antenna can be correctly pointed in the intended directions. These results verify the 2-D wavefront engineering of the 2-D metasurface antenna.

We have finished the design of the 2-D metasurface antenna. Experimental results will be given in a future paper when the measurement results are available. The contents for extending to a 2-D metasurface antenna have been added as Supplementary Note 4 of the revised manuscript.

Comment:

3. Given the matters and problems mentioned above the last sentence stating that “the proposed concept can be extended to the terahertz and optical ranges, leading to widespread applications such as augmented reality, holography, integrated sensing and communications for 6G, quantum optics, and quantum information science” should be either deleted or backed up with considerations of how this extending “to the terahertz and optical ranges” can be implemented in practice, since the latter is very far from being straightforward. Note that the demonstrated transmission data rates of 1 and 2 Mbps are far too low for the information communication systems that are already in use, making it difficult to share enthusiasm of the authors with respect to the potential of the proposed metasurface configuration for future use.

Response:

Thank you very much for your valuable comments. In the terahertz band, the ON-OFF switching of the meta-atom can be implemented using a complementary metal-oxide-semiconductor (CMOS)-based chip [R15]. In fact, we have recently designed a space-time-coding (STC) antenna array operating at around 300 GHz based on the CMOS integrated circuit (IC). The layout and photograph of the prototype under a microscope are shown in Figs. R12a and b, respectively. The terahertz STC IC consists of 2×4 -elements and can be scalable to larger arrays. We leveraged the subwavelength n-type metal-oxide-semiconductor (NMOS) transistor as the terahertz switch to switch the radiating states of each slot antenna element. The terahertz IC was fabricated by Taiwan Semiconductor Manufacturing Company Limited (TSMC) based on the 65-nm industry-standard CMOS process. The terahertz STC IC was measured in the State Key Laboratory of Terahertz and Millimeter Waves at the City University of Hong Kong, as shown in Fig. R12c. Our preliminary measurement results show that the terahertz STC IC can realize direct information transmission with OOK and BPSK modulation schemes at 308 GHz, as shown in Figs. R12d,e. More measurements of the terahertz IC are undergoing. After getting all the measurement results and paper materials ready, we will be pleased to share with everybody this exciting work in the near future.

Fig. R12 | Terahertz spatiotemporally modulated antenna array. **a-b**, Layout and photograph of the IC prototype. **c**, Measurement setup. **d-e**, Measured constellation diagram and eye diagram at the receiver end as the STC IC operates in BPSK (d) and OOK (e) modulation scheme.

Our preliminary results and the programmable terahertz holographic metasurface in ref. [R15] verify that the CMOS chip is a promising solution to extend our universal metasurface antenna into the terahertz band. In addition to the CMOS chip, other high-speed switches, such as monolayer molybdenum disulfide switch [R16] and high-electron-mobility InAlN/GaN-based metadvice switch [R17], are also reported for the terahertz frequency band.

The nanoscale optical switch can be implemented in the optical band by phase-changed materials [R18]-[R19]. The phase-changed material based on germanium antimony telluride (GST) can be high-speed electrically switched between the amorphous and crystalline states. Electrical tuning of the metasurface and antenna has been demonstrated [R19].

As per your suggestion, we have provided the promising technologies that can extend the proposed universal metasurface antenna into terahertz and optical bands on Page 14 of the revised manuscript.

[R15] Venkatesh, S., Lu, X., Saeidi, H. & Sengupta, K. A high-speed programmable and scalable terahertz holographic metasurface based on tiled CMOS chips. *Nat. Electron.* 3, 785-793 (2020).

[R16] Kim, M. et al. Monolayer molybdenum disulfide switches for 6G communication systems. *Nat. Electron.* 5, 367-373 (2022).

[R17] Samizadeh Nikoo, M. & Matioli, E. Electronic metadevices for terahertz applications. *Nature* 614, 451-455 (2023).

[R18] Zhang, Y. et al. Electrically reconfigurable non-volatile metasurface using low-loss optical phase-change material. *Nat. Nanotechnol.* 16, 661-666 (2021).

[R19] Wang, Y. et al. Electrical tuning of phase-change antennas and metasurfaces. *Nat. Nanotechnol.* 16, 667-672 (2021).

As in our reply to the comments from the Reviewer #1, the transmission data rate in this proof-of-concept is mainly limited by the low-speed interconnector (Fig. R13a) between the FPGA board and metasurface antenna board. The adopted PIN diode (MACOMMADP-000907-14020x) can support the fastest switching speed of 2-3 ns. Although the adopted FPGA (ALTERA Cyclone IV) allows a maximum internal clock speed of 1.3 GHz using a phase-locked loop, the relatively low-speed interconnector (Fig. R13a) between the FPGA board and metasurface antenna board limits the maximum control speed of the whole metasurface system. Figs. R13b-d show the measured control signal waveforms with different frequencies from the FPGA using an oscilloscope. We can observe that the signal waveform with 50 MHz distorts from a perfect rectangular waveform due to the parasitic inductance of the interconnector. Therefore, the maximum frequency shift for the current metasurface system is around 20 MHz and the transmission data rate of 2 Mbps for information transmission.

Fig. R13. | **a**, Photograph of the interconnect between the FPGA board and metasurface antenna board. **b-c**, Measured control signal waveforms from FPGA with frequencies of 10 MHz (b), 20 MHz (c), and 50 MHz (d) using an oscilloscope.

We want to emphasize that the current universal metasurface antenna is a proof-of-concept example demonstrating its flexibility for complete control of the five fundamental wave properties and direct information transmission, although not at a high-speed data rate. Furthermore, the limitation of the control speed of the metasurface system can be largely mitigated by adopting, e.g., 1) high-speed interconnect technologies (such as FMC connector, PCI Express), 2) packaging the FPGA and metasurface on the same PCB board, and 3) using high-speed PCB routing.

This information has been added to Pages 21-22 of the revised manuscript.

Comment:

4. Minor issues: (a) the references used in the first sentence “Electromagnetic (EM) waves, ranging from microwave and terahertz waves to visible light, are the bases of various disciplines ranging from optics ...” seem to be out place, being not general (e.g., not about optics as such) but very specifically dealing with metasurfaces and metamaterials; (b) “universal metasurface antenna” is found in 71 places of the main text, meriting definitely to abbreviate, while other abbreviations should be checked whether their use is reasonable or not (using 2-3 times is not sufficient, in my opinion).

Response:

Thank you very much for your suggestion. The references in the first sentence have

been updated with more general EM-related references in the revised manuscript.

As per your suggestion, the abbreviation “UMA” has been utilized to replace “universal metasurface antenna” in the main text and Supplementary Information. Other abbreviations only used 2-3 times, such as STMM (spatiotemporally modulated metasurface), LiDAR (light detection and ranging), PMC (phase matching condition), radiofrequency (RF), have been deleted in the revised manuscript.

REVIEWER COMMENTS

Reviewer #1 (Remarks to the Author):

Although there are still some concerns (e.g., limitations, speed, claims on THz and optical regions, etc.), the manuscript now includes important details along with the strengths and weaknesses. It may be a useful resource (along with the referee comments) and have some impact on future research in this direction to address those concerns.

Reviewer #3 (Remarks to the Author):

The authors have introduced significant modifications to the manuscript, following comments from the reviewers, and thereby substantially improved the paper.

I am however not satisfied with their response on my comment 2 regarding the demonstration of the wave focusing shown in Fig. 3g. The absence of the focusing effect in the experimental results, which are shown in Fig. 3g, could have been related to a very large beam divergence out of the considered x-z plane as I suggested in my comment. The authors have now demonstrated by simulations that the divergence effect (and the fading associated with that) alone does not lead to the disappearance of the focusing effect (Figs. S14 & S15).

In the main text, the authors have added one sentence "The effects of the 1D and 2D radiating aperture on the fading distance for the Airy beam and focusing beam are investigated in Supplementary Note 6". This sentence along with the new material and Figs. S14 & S15, which are added to the Supplementary Information" is absolutely insufficient, because it does not explain the absence of the focusing effect as visually perceived.

None of the 5 panels in Fig. 3g shows a visually clear focusing effect. Moreover, their appearance is in the stark contrast with those of the figures shown by the authors in their reply: the focusing effect is very clearly seen in Figs. S14 & S15. In fact, these figures, by demonstrating a very clear focusing effect (i.e., showing a beam decreasing in width and increasing in the intensity when approaching the focal plane), emphasize poor performance (as far as the focusing effect is concerned) of the universal metasurface antenna.

Overall, I reiterate my previous critic and insist on introducing the appropriate discussion of the experimental results shown in Fig. 3g. The authors do illustrate with Fig. 3f that the "focused" beam is displaced laterally along the x-axis when scanning the focal spot. Similarly, it would be appropriate to show cross sections along the beam propagation to clearly reveal the increase in the intensity at (or close to) the designed distance of 80 mm. In the absence of a clear visual focusing effect in Fig. 3g, this additional figure is mandatory, in my opinion.

Responses to Reviewers' Comments

The authors would like to thank the editors and reviewers for their time and constructive comments to improve the manuscript. The replies to all comments are provided below in blue colors. The revised portions in the manuscript are highlighted in yellow background.

Responses to Reviewer #1:

Comment:

Although there are still some concerns (e.g., limitations, speed, claims on THz and optical regions, etc.), the manuscript now includes important details along with the strengths and weaknesses. It may be a useful resource (along with the referee comments) and have some impact on future research in this direction to address those concerns.

Response:

We thank the referee for the positive feedback and the kind support.

Responses to Reviewer #3

Comment:

The authors have introduced significant modifications to the manuscript, following comments from the reviewers, and thereby substantially improved the paper.

I am however not satisfied with their response on my comment 2 regarding the demonstration of the wave focusing shown in Fig. 3g. The absence of the focusing effect in the experimental results, which are shown in Fig. 3g, could have been related to a very large beam divergence out of the considered x-z plane as I suggested in my comment. The authors have now demonstrated by simulations that the divergence effect (and the fading associated with that) alone does not lead to the disappearance of the focusing effect (Figs. S14 & S15).

In the main text, the authors have added one sentence “The effects of the 1D and 2D radiating aperture on the fading distance for the Airy beam and focusing beam are investigated in Supplementary Note 6”. This sentence along with the new material and Figs. S14 & S15, which are added to the Supplementary Information” is absolutely insufficient, because it does not explain the absence of the focusing effect as visually perceived.

None of the 5 panels in Fig. 3g shows a visually clear focusing effect. Moreover, their appearance is in the stark contrast with those of the figures shown by the authors in their reply: the focusing effect is very clearly seen in Figs. S14 & S15. In fact, these figures, by demonstrating a very clear focusing effect (i.e., showing a beam decreasing in width and increasing in the intensity when approaching the focal plane), emphasize poor performance (as far as the focusing effect is concerned) of the universal metasurface antenna.

Overall, I reiterate my previous critic and insist on introducing the appropriate discussion of the experimental results shown in Fig. 3g. The authors do illustrate with Fig. 3f that the “focused” beam is displaced laterally along the x-axis when scanning the focal spot. Similarly, it would be appropriate to show cross sections along the beam propagation to clearly reveal the increase in the intensity at (or close to) the designed distance of 80 mm. In the absence of a clear visual focusing effect in Fig. 3g, this additional figure is mandatory, in my opinion.

Response:

Thank you very much for your constructive comments, which are very important to improve our manuscript. We have further measured the E-field intensity distributions

when the designed focus point is located at $F(x_F, y_F, z_F) = (0, 0, 40)$ mm, $(0, 0, 60)$ mm, $(0, 0, 80)$ mm, as shown in Fig. R1. It can be observed that a smaller depth of focus (DoF) with a clear visual focusing effect is achieved as the designed focal point is closer to the universal metasurface antenna. The same conclusion is obtained for near-field focused planar array antennas (see Fig. 5 in Ref. [R1]). This is reasonable as the focusing effect is weaker when the designed focus moves away from the radiating aperture. The near-field focused effect disappears as the designed focal point shifts to infinite; in this case, the near-field focused antenna degenerates into a far-field high-directivity antenna with an equal aperture phase distribution.

Fig. R1. Measured E-field intensity distributions at the $m = -1$ harmonic frequency for the designed focus point at $F(x_F, y_F, z_F) = (0, 0, 40)$ mm, $(0, 0, 60)$ mm, and $(0, 0, 80)$ mm, respectively.

Due to the clear visual focusing effect for a closer focal point, we take the focus scanning on the focal plane $z = 60$ mm as an illustrative example in the revised manuscript. The results are shown in Fig. R2a, demonstrating the universal metasurface antenna's lateral focus steering capability. As per your suggestion, Fig. R3 shows the measured cross-section intensity distribution in different transversal planes for the designed focal point $F = (0, 0, 60)$ mm. We can observe that the intensity increases as close to the focal plane. Nevertheless, the density peak occurs at around $z = 50$ mm. It is well-known that for near-field focused antennas, the power-density peak is not located at the designed focal point (the position where all the radiated fields interfere constructively) [R1-R3]. Due to the free-space wave spreading factor $1/R$, the density peak is actually located between the designed focal point and the antenna's aperture [R1-R3]. The discrepancy between the intensity peak and designed focal point position is more obvious for off-axis focusing scenarios, as shown in Fig. R2a. However, we would like to highlight that the strongest field on the interested focal plane $z = 60$ mm is exactly located at the designed focal position for all the focusing scenarios (see Fig. R2b, which shows the 1-D normalized intensity distributions on the focal plane). The target/sample is placed on the focal plane [R4] for microwave imaging applications.

Therefore, the well-defined focused beam on the focal plane generated by our universal metasurface antenna can be used for real-time sensing and detection.

Fig. R2. **a**, Measured E-field intensity distributions at the $m = -1$ harmonic frequency for different focus spots with $F_1 = (-60, 0, 60)$ mm, $F_2 = (-30, 0, 60)$ mm, $F_3 = (0, 0, 60)$ mm, $F_4 = (30, 0, 60)$ mm, and $F_5 = (60, 0, 60)$ mm. **b**, The corresponding normalized intensity distributions on the focal plane $z = 60$ mm.

Fig. R3. Measured cross-section intensity distributions at the $m = -1$ harmonic frequency in different transversal planes for the designed focal point $F = (0, 0, 60)$ mm.

The measured 1-D intensity distributions along the propagation direction and on the focal plane have been added as Figs. 3f and 3h of the revised manuscript. The phenomenon of a weaker focusing effect for a larger focus distance has been discussed in the revised manuscript and Supplementary Note 6 of the revised Supporting

Information.

References

- [R1] Buffi, A., Nepa, P. & Manara, G. Design criteria for near-field-focused planar arrays. *IEEE Antennas Propag. Mag.* 54, 40-50 (2012).
- [R2] Karimkashi, S. & Kishk, A. A. Focusing properties of Fresnel zone plate lens antennas in the near-field region. *IEEE Trans. Antennas Propag.* 59, 1481-1487 (2011).
- [R3] Li, P.-F., Qu, S.-W., Yang, S. & Nie, Z.-P. Microstrip array antenna with 2-D steerable focus in near-field region. *IEEE Trans. Antennas Propag.* 65, 4607-4617 (2017).
- [R4] Mutai, K. K., Sato, H. & Chen, Q. Active millimeter wave imaging using leaky-wave focusing antenna. *IEEE Trans. Antennas Propag.* 70, 3789-3798 (2021).

REVIEWERS' COMMENTS

Reviewer #3 (Remarks to the Author):

I don't require further changes.

Responses to Reviewers' Comments

Responses to Reviewer #3:

Comment:

I don't require further changes.

Response:

We thank the referee for the positive feedback and the kind support.